# 3D-CoS: A New 3D Reconstruction Paradigm Based on VLM Code Synthesis

## Abstract

Most recent 3D reconstruction and editing systems operate on implicit and explicit representations such as NeRF, point clouds, or meshes. While these representations enable high-fidelity rendering, they are inherently low-level and hard to control automatically. In contrast, we advocate a new *3D* reconstruction paradigm based on vision-language models (VLMs) *Co*de *S*ynthesis (*3D-CoS*), where 3D assets are constructed as executable Blender code, a programmatic and interpretable medium. To assess how well current VLMs can use code to represent 3D objects, we evaluate leading open-source and closed-source VLMs in code-based reconstruction under a unified protocol. We further introduce advanced prompting strategies including a planning stage that produces a ratio-based, part-level blueprint before code synthesis, Retrieval-Augmented Generation (RAG) over well-organized Blender API documents, and in-context learning with geometric demonstrations. To demonstrate the unique advantages of this representation, we also present an evaluation focused on localized, text-driven modifications, comparing our code-based edits to state-of-the-art mesh-editing methods. Our study shows that code as a 3D representation offers strong controllability and locality, exhibiting significant advantages in edit fidelity, identity preservation, and overall visual quality. Our work also analyzes the potential of this paradigm and specifically delineates the current capability frontier of VLMs for programmatic 3D modeling, demonstrating the promising future of reconstruction by code.

## 1 Introduction

Recent breakthroughs in foundation models, particularly large vision-language models (VLMs), have led to remarkable progress in multimodal understanding, logical reasoning, and tool usage. These models have shown the ability to operate within a "perception–reasoning–planning–execution" loop, and automatically generate executable code to accomplish complex tasks (Gao et al., 2023; Li et al., 2024b; Liang et al., 2023). This capability suggests a new path for 3D: instead of recovering geometry purely as meshes, point clouds, or implicit fields, we can generate executable programs that reconstruct 3D assets inside 3D engines (e.g., Blender (Blender Online Community, 2025), Unity (Unity, 2025a), Unreal (Epic, 2025b)). Code as a 3D representation brings interpretability, editability, and compositional control. 3D components are constructed in an explicit and parameterized manner, and are verifiable by execution. Besides, 3D engines provide mature API (Blender Foundation, 2025; Epic, 2025a; Unity, 2025b), which further makes programmatic creation, editing, and rendering first-class citizens, providing a practical substrate for automation (Ahuja & Contributors, 2025).

Several recent works have explored the feasibility of using code to generate and edit 3D assets, and two recent lines of work motivate our study. LL3M (Lu et al., 2025) demonstrates text-driven 3D asset creation by coordinating agents that write Blender scripts, evidencing that code can serve as a powerful representation for modeling geometry, layout, and appearance. BlenderGym (Gu et al., 2025) introduces a benchmark that tasks VLM systems with code-D components are constbased 3D scene editing and shows that state-of-the-art models can comprehend programmatic code and further make targeted code-level modifications. Together, these works validate the feasibility of "code for 3D" while revealing a gap in image-conditioned reconstruction and in systematic evaluation specific to 3D reconstruction. In reconstruction, the image input is essential: it supplies silhouette constraints, object pose, and disambiguates topology and fine details that text alone can-

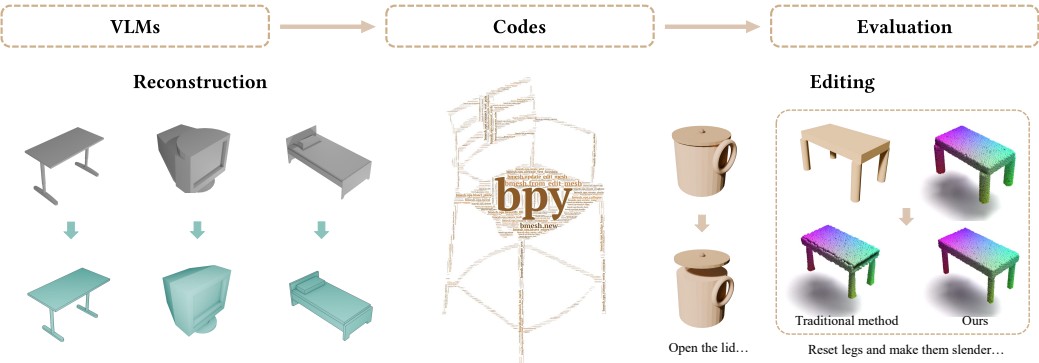

Figure 1: **An overview of our 3D code modeling paradigm.** The top workflow summarizes our core process: code synthesis via VLMs, and its subsequent evaluation. Our work treats code as a unified representation for 3D assets. (Left) We demonstrate its capability in **reconstruction**, generating high-fidelity objects from a single **image**. (Right) We highlight its advantages in **editing**, where code-driven edits achieve superior fidelity compared to traditional methods.

not specify. Equally important is a standardized pipeline to evaluate code-as-representation under image conditioning. We fill this gap with a systematic study of image-conditioned, code-based 3D reconstruction, accompanied by a unified evaluation protocol across multiple VLM families.

We focus on the image → code → 3D setting and ask two questions: Why use code as the 3D representation? Where is the ceiling for code-based reconstruction? For the first, code is a high-level, structured, parameterized medium that enables fine-grained control and reliable iteration—advantages that are hard to obtain with purely implicit (e.g., NeRF (Mildenhall et al., 2021), 3D Gaussian Splatting (Kerbl et al., 2023)) or low-level explicit (e.g., mesh, point cloud) forms. Besides, mature ecosystems such as Blender, with its comprehensive Python API (Blender Foundation, 2025), enable programmable creation and editing of objects, providing a solid interface foundation for automation. For the second, a significant portion of foundational 3D resources are inherently programmatic: ModelNet (Wu et al., 2015) and ShapeNet (Chang et al., 2015) organize large collections of Computer-aided design (CAD) models; the Fusion 360 Gallery captures programmatic parametric CAD by logging human sketch-and-extrude timelines, and it also releases a reconstruction set of 8,625 design sequences (Willis et al., 2021). If we target "recovering an executable modeling program", the representational ceiling can at least reach parametric programmatic modeling created by humans.

In this work, we propose a novel paradigm for 3D reconstruction using programmatic code on the Blender platform (Figure 1). We demonstrate its advantages in terms of editability, control granularity, and interpretability compared to other mesh representations. To systematically evaluate the capabilities of modern VLMs in this setting, we introduce a code-based reconstruction benchmark that evaluates state-of-the-art open- and closed-source VLMs on single-image reconstruction under unified prompting, and compares them to mesh-based 3D reconstruction baselines. Our evaluation includes 3D metrics to measure geometric similarity and 2D metrics to account for occlusion relationships that 3D metrics ignore, and tests the performance under multi-view observations. To address potential misalignments in pose and scale between generated code-based models and ground truth, we propose a robust registration protocol. Furthermore, we introduce a reconstruction variant with an edit intent prompt and demonstrate its effect. Beyond reconstruction, we include a code-based editing protocol to expose the unique strengths of programmatic control (e.g., targeted parameter changes, retention of unedited areas) relative to mesh-only pipelines, further validating the great potential of the code-based representation paradigm.

Our main contributions are threefold:

- We propose a novel paradigm for 3D reconstruction using Blender Python code, analyzing the potential of this paradigm;

- We construct a reproducible benchmark and metrics suite for Blender-code reconstruction, systematically evaluate the capabilities of state-of-the-art VLMs on the task of code-based 3D reconstruction, and analyze the impact of different prompting strategies;

- We demonstrate the significant advantages of our code-based paradigm for editing tasks, validating its superiority over traditional representations through experimental evaluations.

## 2 RELATED WORK

**Classic 3D Reconstruction Representations.** Existing methods mainly develop along two lines: (i) *Implicit shape representations* such as neural radiance fields based methods (Mildenhall et al., 2021; Poole et al., 2022; Kosiorek et al., 2021; Wang et al., 2023; 2022), 3D Gaussians based methods (Kerbl et al., 2023; Chen et al., 2024; Yi et al., 2024; Wu et al., 2025), and other approaches that learn a latent space and decode it into implicit representations (Zhang et al., 2023; Jun & Nichol, 2023; Lan et al., 2024). This family excels in multi-view consistency and visual fidelity, but typically offers limited precise control, lacks interoperability with standard graphics pipelines, and often relies on heavy optimization or bespoke training. (ii) *Explicit shape representations* (point clouds, voxels, meshes) are more amenable to geometric measurement and integration with existing engines, and have been extensively studied (Chen et al., 2021; Li et al., 2021; Ibing et al., 2021; Vahdat et al., 2022). However, they operate at a lower semantic level: mesh/point-cloud vertices and faces are the consequences, rather than the intent of modeling. They lack shared high-level primitives and constraints, making automated control and cross-category, generalizable editing challenging.

Therefore, to jointly pursue interpretability, controllability, and engineering deployability, we advocate using Blender Python code as a unified representation of 3D objects. Its modeling primitives and operators (e.g., primitive_cylinder_add, bevel) naturally carry human modeling semantics, supporting modularity and compositionality. By editing code, one can readily modify an object's geometry and texture, enabling precise control over the resulting mesh and making this representation well suited for automated 3D workflows.

**Code Based 3D Representations.** Beyond implicit and explicit geometry, another line of work represents shapes as programs. In the direction of general domain-specific language (DSL) methods, 3D Shape Programs (Tian et al., 2019) encode repeated and symmetric structures as programs. CSGNet (Sharma et al., 2018) parses 2D and 3D shapes into CSG programs, demonstrating the compactness and interpretability of programmatic representations. ShapeAssembly (Jones et al., 2020) and subsequent work such as ShapeMOD (Jones et al., 2021) design DSLs specifically for 3D shape structures, constructing hierarchical and reconfigurable shape programs. These approaches often rely on custom DSLs whose geometric resolution is limited and which are not tightly integrated with mainstream graphics engines. In the CAD-oriented line of code-based representations, methods such as DeepCAD (Wu et al., 2021) treat CAD operation sequences (sketch, extrude, etc.) as program sequences. Later text- and point-to-CAD methods (Xu et al., 2024b) similarly exploit parametric operation sequences to obtain editable designs, but remain tied to specific CAD environments.

These methods either use custom DSLs or commercial CAD environments, typically rely on bespoke languages or closed modeling stacks, and often abstract away from the high-fidelity meshes and materials used in production. In contrast, we adopt Blender Python as a unified, executable code representation. Our models operate directly over the native primitives and operators (`primitive_cylinder_add`, `bevel`), so that the inferred programs are immediately compatible with standard 3D workflows and can be precisely edited, rendered, and deployed.

**Large Models for 3D Generation and Editing via Code.** The success of Large Models (LMs) in leveraging code to solve problems (Gao et al., 2023) has inspired exploration into using LMs to generate code for manipulating 3D objects. BlenderAlchemy (Huang et al., 2024) generates materials in Blender for existing geometry. SceneCraft (Hu et al., 2024) retrieves 3D assets and employs an LLM to organize them into a coherent spatial layout. 3D-GPT (Sun et al., 2025) produces parameters for Infinigen (Raistrick et al., 2023), a pre-existing procedural generator specializing in predefined scenes, particularly natural environments. LL3M (Lu et al., 2025) generates 3D assets from text guidance, incorporating geometry and appearance attributes and BlenderMCP (Ahuja & Contributors, 2025) uses a single LLM calling Blender functions via the Model Context Protocol (Anthropic PBC, 2024). BlenderGym (Gu et al., 2025) utilizes VLMs for 3D scene reconstruction through code editing. MeshCoder (Dai et al., 2025) fine-tunes an LLM to translate 3D point clouds into editable Blender scripts. However, these methods either do not emphasize the unique advantages of

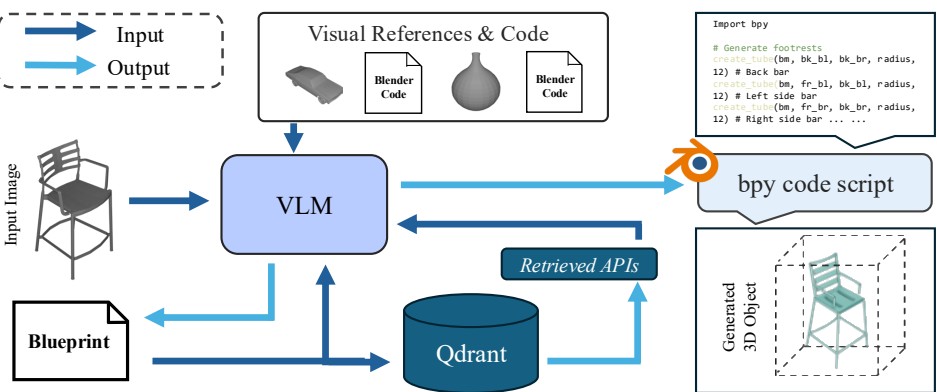

Figure 2: **Overview of our VLM-based 3D reconstruction pipeline.** Given a single input image, the VLM, depending on the paradigm, derives a quantitative blueprint, augmented with visual code references and/or APIs retrieved from Blender documentation, to synthesize a bpy script whose execution produces a 3D object matching the input image. Full versions of the blueprint/APIs/code are provided in Appendix A.6.

code over traditional 3D representations, or don't tackle image-conditioned 3D reconstruction from natural images.

In contrast, we demonstrate the benefits of code compared to traditional mesh-based representations and provide a comprehensive evaluation of the capability of current VLMs to reconstruct 3D objects from image inputs.

## 3 METHOD

### 3.1 3D RECONSTRUCTION PARADIGMS FOR CODE SYNTHESIS

**Problem setup.** Given a single input image $\mathcal{I}$, the goal is to produce an executable programmatic script that reconstructs geometry such that the rendered result matches $\mathcal{I}$ as closely as possible. To isolate the effect of prompting paradigms on geometric faithfulness, we freeze VLMs and vary only the prompting workflow and whether external knowledge is injected. We evaluate four pipelines: *Single-call*, *Planning*, *RAG*, and *Few-shot*, along with a text-conditional reconstruction variant.

**Single-call Paradigm.** We issue a single-turn instruction asking the VLM to reconstruct the object from an input image and return a complete script. The system prompt emphasizes geometry and enforces all logic encapsulated in one callable function with explicit parameters as input. This setting is the shortest and fastest to generate a Blender script but tends to miss details or mis-specify parameters for complex geometry.

**Planning Paradigm.** We decouple *structural perception* from *API utilization* via two stages: Stage 1 extracts a quantitative blueprint from $\mathcal{I}$; Stage 2 treats that blueprint as a supplementary source of geometric truth and synthesizes a Blender script.

*Stage 1: quantitative blueprint.* We instruct the VLM to perform *base-dimension estimation* and construct a blueprint $\mathcal{B}$: pick a single object-level reference size and set it to 1.0 (e.g., overall width/height). All sizes, angles, and hierarchical parameters must then be expressed as ratios with respect to this previously defined base. Besides, we require a *physical feasibility check* that under occlusion or perspective ambiguity, the model must propose minor modifications to ensure structural stability. An example of the blueprint can be found in Section A.6.

*Stage 2: code generation.* Conditioned on $(\mathcal{I}, \mathcal{B})$, we prompt the VLM under the guideline that the script should preserve the numeric structure in $\mathcal{B}$, refer to image only when encountering non-parametric details like complex curves, and only minimal tweaks are allowed to prevent physical impossibility. Compared to Single-call, the Planning pipeline markedly improves interpretability by explicitly providing a parameterized blueprint and separating "what to build" from "how to call the API".

**RAG Paradigm.** Relying on model in-built knowledge alone biases generation toward high-frequency Blender Python APIs seen during training, neglecting more suitable but long-tail APIs. To this end, we convert the Blender 4.4 documentation into a searchable knowledge base and provide the most relevant, complete API candidates to the model as background knowledge during code synthesis.

*Doc-to-database.* Using the Sphinx inventory (The Sphinx Project) and sphobjinv (Skinn, 2024), we crawl and parse documentation pages, extracting callable functions, signatures, and parameter semantics from the `bpy`, `bmesh`, and `mathutils` modules. Each entry is stored under a unified JSON schema as a database item $\mathcal{D}$ with standardized keys, as shown in Figure 11. The resulting database contains 1,683 pages with 21,102 functions. We embed entries and index them in Qdrant (Qdrant Team, 2025) for hybrid (semantic + keyword) retrieval.

*Component-level query generation.* Given the components in $\mathcal{B}$, the VLM generates a query per component with *module preferences* and *keywords* constraints to maximize recall while aligning with the blueprint semantics. The resulting queries express high-level component names as simple geometric shapes (e.g., "chair base plate" $\rightarrow$ "plane") and define key operations (e.g., bevel, solidify), as shown in Figure 12.

*Retrieval and refinement.* In Qdrant we retrieve Top-$k$ ($k = 8$) documentation chunks by hybrid hits, and instruct the VLM to consolidate them into background knowledge $\mathcal{K}$ in a *component $\rightarrow$ candidate-API list* schema. A retrieval result is exhibited in Figure 13.

*Knowledge-injected synthesis.* Final code generation conditions on $(\mathcal{I}, \mathcal{B}, \mathcal{K})$: $\mathcal{B}$ supplies explicit parameters, the image informs non-parametric details and the API function $\mathcal{K}$ obtained by retrieving provides auxiliary knowledge to the model. Retrieval expands the candidate space from high-frequency to *full* documentation coverage, especially long-tail ones.

**Few-shot Paradigm.** To investigate the in-context learning capabilities of VLMs, we prompt the model with a concatenated input consisting of a few exemplary pairs $(\mathcal{I}_{ex}, \mathcal{S}_{ex})$ and a suitable task instruction. Specifically, we utilize *Gemini 3.0 Pro* to generate these scripts on held-out samples disjoint from ModelNet10, refining them under human guidance. We meticulously construct these examples such that the scripts $\mathcal{S}_{ex}$ contain quantitative blueprint logic and correct API usage. By observing these demonstrations, the model is guided to implicitly learn the underlying geometric reasoning patterns and syntactic standards, applying them to the target reconstruction task.

**Variant: From Reconstruction to Editing.** This variant extends reconstruction to incorporate an edit intent specified by text ($\mathcal{T}_{edit}$), producing an edited 3D object from a source image $\mathcal{I}$. This process reuses our Planning workflow. Stage 1 predicts an edited blueprint $\mathcal{B}_{edit}$ by jointly interpreting $(\mathcal{I}, \mathcal{T}_{edit})$. Stage 2 then synthesizes the final programmatic script conditioned on $(\mathcal{I}, \mathcal{B}_{edit})$, preserving the numeric structure. This variant demonstrates the flexibility of our paradigm by unifying reconstruction and editing into a single, conditional generation process.

## 3.2 3D Editing paradigm with Code Modification

A key reason we choose code as the representation for 3D shapes is the flexibility and convenience it provides for subsequent editing operations. When a 3D object generated by code needs to be modified, we can make adjustments directly at the code level, leveraging the VLM's powerful comprehension and reasoning capabilities.

**Code-based 3D Editing Paradigm.** This modality is designed for the editing of existing programmatic 3D assets. The inputs are a source *bpy* script ($\mathcal{S}_{src}$) and a textual edit instruction ($\mathcal{T}_{edit}$). In this paradigm, the source script serves as a complete and structured description of the initial 3D model. The VLM's core task here is comprehension and transformation: it must first parse the logic of $\mathcal{S}_{src}$ to identify the code segments corresponding to the instruction, then precisely modify that segment according to the prompt and output a new, edited *bpy* script ($\mathcal{S}_{dst}$).

**Construction of Localized 3D Editing Assets.** We build upon the *BlendNet* dataset (Du et al., 2024), which contains pairs of *bpy* code and corresponding textual descriptions. From this dataset, we select 55 representative samples covering a diverse range of objects. We then manually authored a high-quality and specific editing instruction for each sample. This process results in a new dataset, *BlendNet-E*, where each entry is a triplet: (source script$_i$, source description$_i$, edit instruction$_i$).

## 3.3 EVALUATION

To assess the current capability frontier of VLMs for programmatic 3D modeling, we designed a comprehensive evaluation protocol, including spatial registration of 3D objects, dedicated evaluation datasets, and a suite of 3D and 2D metrics to quantify their reconstruction quality.

**3D Model Registration.** The objects synthesized by code may lack the absolute scale, position, and orientation compared to the ground truth object. Therefore, a robust registration step is required before the quantitative comparison. The protocol first normalizes the scale of the generated 3D object to the scale of the ground truth 3D object. Subsequently, we employ a coarse-to-fine alignment strategy to find the optimal rigid transformation, leveraging a RANSAC-based algorithm to match the Fast Point Feature Histograms (FPFH) (Rusu et al., 2009) and a point-to-plane Iterative Closest Point (ICP) algorithm to minimize the final alignment error. The resulting transformation matrix is then applied to the generated mesh.

**Datasets for Reconstruction.** We use the ModelNet10 (Wu et al., 2015) dataset and follow a controlled rendering protocol: each object is normalized to unit length and rendered from eight viewpoints evenly distributed on a sphere with a radius of 1.76. We also produce depth and normal maps for analysis. A human annotator selects the most informative RGB view among the eight as the input image to the reconstruction pipeline. Besides, ModelNet10 is split into ModelNet10-*easy* and ModelNet10-*hard* parts based on 3D object structure complexity with one human annotator and one human verifier. Details of splits information can be found in Section A.2.

**Dataset for Reconstruction Variant.** To evaluate the text-conditional reconstruction variant, we construct a test set derived from the ModelNet10 assets described above. For this new dataset, we use the same human-selected "most informative" rendered views as the image inputs. For each input image, we prompt GPT-4o (OpenAI, 2024) to generate a high-level editing instruction tailored to the object depicted. This process results in a dataset of 100 triplets, each of which contains a source rendered view, a synthetic editing instruction, and the ground truth `.blend` file for the original object. We refer to this new dataset as *ModelNet10-V*.

## 4 EXPERIMENTS

### 4.1 3D RECONSTRUCTION EVALUATION

Our goal is to generate executable *bpy* code from a single input image and render a 3D object that matches the target as closely as possible. We evaluate four prompting paradigms aforementioned in Section 3.1 with datasets constructed in Section 3.3.

#### 4.1.1 EXPERIMENTAL SETUP

**Models.** We evaluate code-based reconstruction on both *open-source* and *closed-source* VLMs, and compare to classical mesh baselines. Open-source VLMs: InternVL3.5-38B (Wang et al., 2025), LLaVA-OneVision-Qwen2-72B (Li et al., 2024a), Qwen2.5-VL-72B-Instruct (Bai et al., 2025). Closed-source VLMs: Claude Sonnet 4.0 (Anthropic PBC, 2025), o3 (OpenAI, 2025), Gemini 2.5 Pro (Google DeepMind, 2025a), Gemini 3.0 Pro (Google DeepMind, 2025b). Classical mesh baselines: Unique3D (Wu et al., 2024) and InstantMesh (Xu et al., 2024a). All models receive the same single RGB view, based on which VLMs emit a Blender script executed in headless Blender 4.4, and classical methods directly reconstruct meshes.

**Metrics.** To conduct a comprehensive evaluation, we employ a specific suite of widely-used 3D and 2D metrics. It should be noted that the evaluations are performed after the 3D model registration described in Section 3.3.

*3D Metrics.* To evaluate the overall 3D shape, we compute three key metrics: (i) the Chamfer Distance (CD), which measures the average closeness between the surfaces of the two models; (ii) the 3D Intersection-over-Union (3D IoU), which assesses volumetric overlap by converting the models to voxels; and (iii) the F-score@5%, which balances accuracy and completeness with a distance threshold of 5% relative to the ground-truth bounding box diagonal.

*2D Metrics.* As a supplement to 3D metrics, 2D metrics explicitly take into account occlusion relationships between components. To assess view-dependent accuracy, we render both models

Table 1: **Reconstruction on ModelNet10.** 3D metrics: CD = Chamfer Distance, 3D IoU = 3D Intersection-over-Union, F@5% = F-score at 5% threshold. 2D metrics: NRMSE = Normalized RMSE, SSIM = Structural Similarity, MAE = Mean Angular Error (normalized to [0, 1]). "Sin." stands for "single-call", "Pla." for "planning", and "Few." for "Few-shot". The best value in each block is highlighted in green, and the second best value in blue.

| Model | Paradigm | ModelNet10 | | | | | |
| | | 3D Metrics | | | 2D Metrics | | |
| | | CD↓ | 3D IoU↑ | F@5%↑ | NRMSE↓ | SSIM↑ | MAE↓ |
| *Traditional baselines* | | | | | | | |
| Unique3D (Wu et al., 2024) | — | 0.0536 | 0.1469 | 0.6311 | 0.0970 | 0.8489 | 0.2191 |
| InstantMesh (Xu et al., 2024a) | — | **0.0218** | **0.3049** | **0.8809** | **0.0597** | **0.9156** | **0.1241** |
| *Open-source VLM families* | | | | | | | |
| LLaVA-OneVision-Qwen2-72B (Li et al., 2024a) | Sin. | 0.0811 | 0.1135 | 0.4631 | 0.1862 | 0.7910 | 0.2375 |
| | Pla. | 0.0565 | 0.1563 | 0.5925 | 0.1480 | 0.8340 | 0.2450 |
| | RAG | 0.0673 | 0.1523 | 0.5669 | 0.1342 | 0.8347 | 0.2282 |
| | Few. | 0.0654 | 0.1307 | 0.5186 | 0.1866 | 0.8224 | 0.2388 |
| InternVL3.5-38B (Wang et al., 2025) | Sin. | 0.0609 | 0.1575 | 0.5901 | 0.1462 | 0.8435 | 0.2263 |
| | Pla. | 0.0545 | 0.1678 | 0.6243 | 0.1207 | 0.8506 | 0.2150 |
| | RAG | 0.0541 | 0.1675 | 0.6280 | 0.1198 | 0.8542 | 0.2062 |
| | Few. | 0.0474 | 0.1807 | 0.6679 | 0.1204 | 0.8594 | 0.2139 |
| Qwen2.5-VL-72B-Instruct (Bai et al., 2025) | Sin. | 0.1730 | 0.1691 | 0.6480 | 0.1308 | 0.8595 | 0.2154 |
| | Pla. | 0.0524 | 0.1858 | 0.6382 | 0.1037 | 0.8658 | **0.1953** |
| | RAG | 0.0472 | 0.2009 | 0.6821 | **0.1071** | 0.8507 | 0.2042 |
| | Few. | **0.0416** | **0.2031** | **0.7194** | 0.1182 | 0.8759 | 0.2699 |
| *Closed-source VLM families* | | | | | | | |
| Claude Sonnet 4.0 (Anthropic PBC, 2025) | Sin. | 0.0348 | 0.2270 | 0.7664 | 0.0975 | 0.8849 | 0.1758 |
| | Pla. | 0.0363 | 0.2314 | 0.7534 | 0.1022 | 0.8811 | 0.1875 |
| | RAG | 0.0345 | 0.2355 | 0.7723 | 0.0954 | 0.8913 | 0.1841 |
| | Few. | 0.0333 | 0.2398 | 0.7833 | 0.0927 | 0.8893 | 0.2217 |
| o3 (OpenAI, 2025) | Sin. | 0.0434 | 0.1838 | 0.7007 | 0.1087 | 0.8693 | 0.2041 |
| | Pla. | 0.0329 | 0.2309 | 0.7955 | 0.0878 | 0.8934 | 0.1602 |
| | RAG | 0.0302 | 0.2564 | 0.8107 | 0.0830 | 0.9012 | 0.1635 |
| | Few. | 0.0283 | 0.2781 | 0.8204 | 0.0834 | 0.9021 | 0.1789 |
| Gemini 2.5 Pro (Google DeepMind, 2025a) | Sin. | 0.0388 | 0.2137 | 0.7269 | 0.1059 | 0.8733 | 0.1900 |
| | Pla. | 0.0285 | 0.2697 | 0.8287 | 0.0807 | 0.9076 | 0.1537 |
| | RAG | 0.0266 | 0.2977 | 0.8626 | 0.0742 | 0.9093 | 0.1530 |
| | Few. | 0.0294 | 0.2691 | 0.8030 | 0.0861 | 0.8992 | 0.1810 |
| Gemini 3.0 Pro (Google DeepMind, 2025b) | Sin. | 0.0322 | 0.2640 | 0.8043 | 0.0845 | 0.8952 | 0.1576 |
| | Pla. | 0.0247 | 0.2938 | 0.8619 | 0.0791 | 0.9130 | 0.1464 |
| | RAG | 0.0245 | 0.3024 | 0.8746 | 0.0770 | 0.9151 | **0.1430** |
| | Few. | **0.0223** | **0.3083** | **0.8824** | **0.0659** | **0.9212** | 0.1492 |

from the same camera angle and compare their appearance. Specifically, we evaluate two aspects: (i) Depth Error and Similarity (using NRMSE and SSIM) to verify the correctness of visible surface structures by comparing depth maps; and (ii) the Mean Angular Error (MAE) to check the accuracy of surface orientation by comparing normal maps.

### 4.1.2 MAIN RESULTS

Table 1 compares the performance of VLMs with different prompting paradigms and mesh-based methods.

**Closed-source models outperform open-source overall.** Comparing peak performances, *Gemini 3.0 Pro–Few-shot* surpasses the best open-source counterpart (*Qwen–Few-shot*) by ∼46% in CD and ∼52% in IoU. We attribute this distinct gap to stronger vision–geometry alignment and superior in-context learning capabilities in closed-source models. Visual comparisons in Figure 3 confirm that while open-source models capture general shapes, closed-source models excel at fine-grained details.

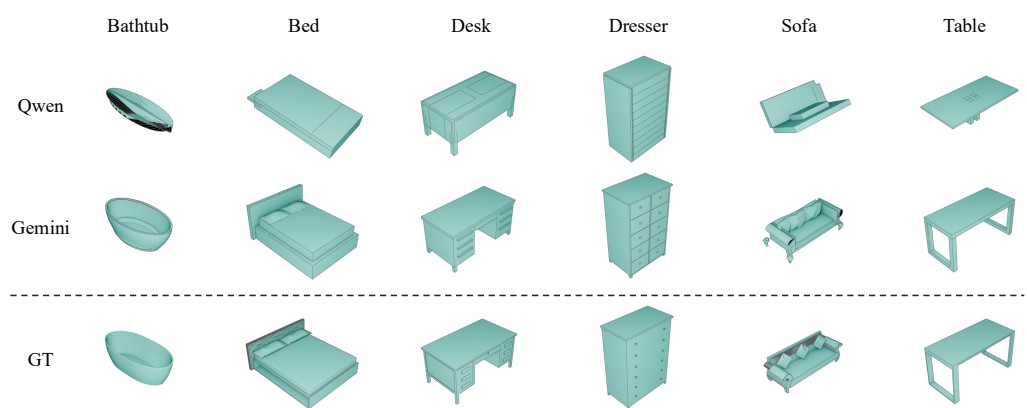

Figure 3: **Code-based reconstruction on ModelNet10.** All inputs come from the ModelNet10 dataset and contain only geometry; the cyan shading is for visualization only.

Table 2: **Qualitative reconstruction results on ModelNet10-*easy/hard* split.** The Planning paradigm is shown for VLMs. Traditional baselines do not use Blueprint/RAG.

| ModelNet10 Split | Unique3D | | | InstantMesh | | | o3 | | | Gemini 3.0 Pro | | |
|---|---|---|---|---|---|---|---|---|---|---|---|---|
| | CD↓ | IoU↑ | F@5%↑ | CD↓ | IoU↑ | F@5%↑ | CD↓ | IoU↑ | F@5%↑ | CD↓ | IoU↑ | F@5%↑ |
| Hard | **0.0515** | **0.1544** | **0.6472** | 0.0219 | 0.2946 | **0.8831** | 0.0359 | 0.2137 | 0.7638 | 0.0266 | 0.2903 | 0.8485 |
| Easy | 0.0557 | 0.1395 | 0.6151 | **0.0217** | **0.3153** | 0.8788 | **0.0298** | **0.2481** | **0.8271** | **0.0227** | **0.2974** | **0.8753** |

**Code-based reconstruction beats Unique3D and rivals InstantMesh.** Our code-based systems significantly outperform Unique3D on the whole dataset, indicating more topologically complete and structurally consistent assemblies. Furthermore, while InstantMesh serves as a formidable baseline, the code-based paradigm has closed the performance gap. Notably, *Gemini 3.0 Pro* with Few-shot sets a new state-of-the-art, achieving 0.3083 in IoU and 0.8824 in F@5%, surpassing *InstantMesh* (0.3049 and 0.8809) in volumetric metrics. This indicates that with sufficient reasoning capacity, VLMs can synthesize geometry with fidelity comparable to or exceeding feed-forward mesh regression, effectively handling the complex structures and high-curvature details that previously challenged weaker models.

**Blueprint helps broadly while RAG helps closed-source more than open-source.** Planning consistently improves over Single-call for most backbones (e.g., InternVL, LLaVA-OneVision, o3, Gemini) by decoupling *"what to build"* from *"how to call APIs"*, thus reducing scale drift and component errors. RAG further boosts closed-source models, but its effect is mixed on open-source: *Qwen* improves, *InternVL* sees minor changes, while *LLaVA-OneVision* degrades. This suggests closed-source models better absorb retrieved API reference and resist distraction from irrelevant snippets, whereas some open-source models exhibit weaker evidence-following under retrieval.

**In-context learning internalizes geometric logic effectively.** The Few-shot paradigm yields competitive results, proving effective for models with strong in-context learning capabilities. As shown in Table 1, for models like *o3* and *Gemini 3.0*, the Few-shot setting achieves comparable or superior performance relative to the RAG paradigm. This suggests that providing high-quality demonstrations allows these models to implicitly *internalize* the quantitative blueprint logic and API distribution. However, this trend is not universal; for models like *Gemini 2.5 Pro*, the performance degrades compared to RAG. We hypothesize that while advanced models benefit from the implicit logic in examples, other models may be distracted by the specific geometric instances in the few-shot context, leading to interference rather than generalization.

**Easier shapes benefit more from code-based pipelines.** As illustrated in Table 2, on ModelNet10–*easy*, closed-source models uniformly outperform their *hard* counterparts. Easy shapes feature fewer parts and more regular configurations, which align naturally with primitives and common modifiers, while hard shapes exhibit curved, non-uniform transitions and fine assemblies that pose more challenges to VLMs.

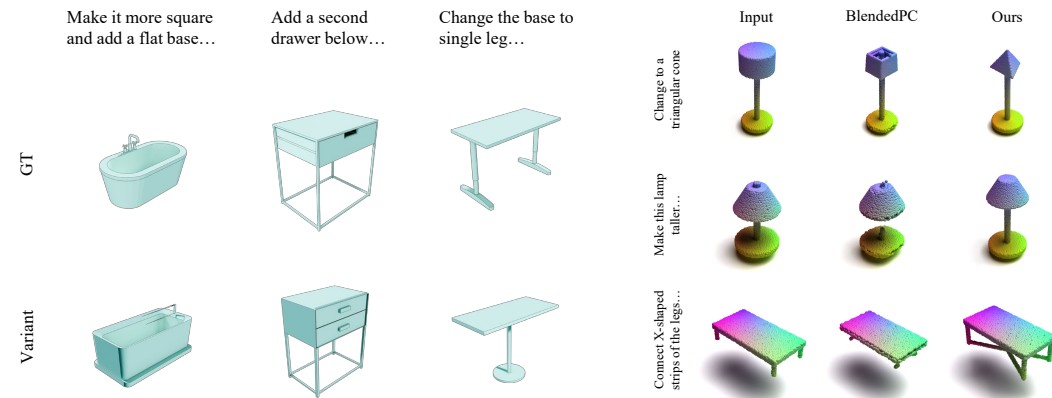

Figure 4: **Reconstruction variant & Code editing results.** (Left) Examples of our text-conditional reconstruction variant, which modifies an object based on a source image and a textual instruction. (Right) A direct comparison of our code editing method against the BlendedPC (Sella et al., 2025) baseline. Code-driven editing demonstrates superior edit fidelity and overall visual quality. A brief instruction is provided for each example in the figure; the full instructions are listed in Section A.5.

| Method | $\text{CLIP}_{sim}$ ↑ | $\text{CLIP}_{dir}$ ↑ |
|--------|-----------|-----------|
| BlendedPC | 0.0142 | 0.2017 |
| $\text{Ours}_P$ | 0.0578 | 0.2499 |
| $\text{Ours}_A$ | 0.0408 | 0.2469 |

| Method | Inst. ↑ | Pres. ↑ |
|--------|---------|---------|
| BlendedPC | 1.90 | 2.45 |
| Ours | 4.37 | 4.30 |

Table 3: CLIP-based similarity and direction scores. $\text{Ours}_P$ is tested on the lamp and table categories compared with BlendedPC, while $\text{Ours}_A$ is tested on the entire *BlendNet-E*.

Table 4: User study comparing BlendedPC and our method. Scores ranges from 1 to 5. "Inst." stands for "Instruction following" and "Pres." stands for "Preservation of unedited regions".

**Reconstruction variants as a bridge to editing.** As an extension of our reconstruction evaluation, we explore the effect of reconstruction with edit intent prompt, which illustrates the flexibility of our paradigm. As presented in the left part of Figure 4, variants are generated by o3, with a single source rendered view from *ModelNet10-V* and corresponding editing instructions. Additional examples are shown in Figure 8. These examples show that our approach is capable of interpreting the textual instruction and applying the corresponding geometric modifications to the object in the source image, showcasing a promising foundation for the code editing paradigm.

## 4.2 3D Code Editing Evaluation

To evaluate the capability of directly editing 3D assets via their code representation, we designed a code editing task, where the model is given a source *bpy* script and a text instruction to modify the object.

### 4.2.1 Experiment Setup

**Dataset.** We evaluate the capability of editing 3D assets via code representation on *BlendNet-E* constructed in Section 3.2.

**Baselines.** We compare our method against BlendedPC (Sella et al., 2025), a state-of-the-art mesh-based editing baseline. We perform an evaluation on a subset of *BlendNet-E* with lamp and table categories as suggested in its code demo. BlendedPC takes point clouds as input. In addition, we evaluate our method with the o3 model across the entire *BlendNet-E* dataset.

**Metrics.** We follow "Edit Fidelity" in BlendedPC (Sella et al., 2025) leveraging the multimodal embedding space of CLIP (Radford et al., 2021) and extend it to multi-view consistency by rendering images from four orthogonal viewpoints. We use the following metrics to evaluate how well the generated results capture the target text cues:

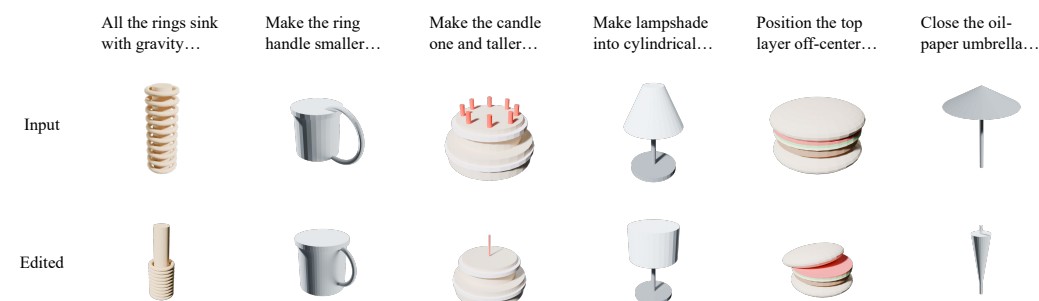

Figure 5: **Qualitative Results for Code Editing.** This figure showcases diverse editing results on objects from the *BlendNet-E* dataset. Each result is shown with a summary of the text instruction used; the complete instructions are detailed in Section A.5.

(i) *CLIP Similarity (CLIP$_{sim}$)*. We measure the cosine similarity between rendered edited objects and their corresponding text descriptions, and report average scores of four views.

(ii) *CLIP Directional Similarity (CLIP$_{dir}$)*. We use the same method as stated in BlendedPC (Sella et al., 2025) to assess whether the edit content is correct.

### 4.2.2 MAIN RESULTS

**Our method consistently outperforms the BlendedPC baseline.** On the lamp/table subset, code-based edit achieves CLIP$_{sim}$ = 0.0578 and CLIP$_{dir}$ = 0.2499, while BlendedPC records 0.0142 and 0.2017, respectively, as shown in Table 3 and in the right part of Figure 4, which is a +3.07× relative increase in similarity and a +23.9% increase in directional consistency, indicating both stronger text-image alignment and more faithful execution of the intended edit. Besides, we conduct a user study on samples generated by BlendedPC. As shown in Table 4, our method shows significant advantages in both "Instruction following" and "Preservation of unedited regions".

**Generalization to other categories.** Evaluated on the full *BlendNet-E* dataset, Ours$_A$ attains CLIP$_{sim}$ and CLIP$_{dir}$ close to the lamp/table result of Ours$_P$, indicating that our edits preserve the intended *semantic direction* across a broader set of shapes. Despite this wider scope, Ours$_A$ remains stably superior to BlendedPC, and achieves a **+187%** gain in CLIP$_{sim}$ and a **+22.4%** gain in CLIP$_{dir}$. Furthermore, Figures 5 and 10 demonstrate that code-based edits effectively implement targeted geometric changes while preserving unmodified parts. This capability highlights the robustness and scalability of programmatic manipulation.

### 4.3 LIMITATIONS

Despite promising results, our work also indicates several limitations. Current VLMs still struggle with fine-grained structural reasoning especially for curved, hierarchical, or interlocking geometries, and exhibit imperfect 3D spatial understanding. Moreover, generating robust programmatic code for complex assemblies remains challenging: models often omit dependencies, break object relationships, or produce non-executable code. Nonetheless, we believe a specially fine-tuned code-centric VLM could substantially improve this.

## 5 CONCLUSION

In this work, we propose and systematically evaluate a new paradigm for 3D reconstruction that treats programmatic code as a bridge between image input and 3D objects, offering significant advantages in interpretability, controllability, and editability over low-level mesh or implicit field representations. We conduct a comprehensive benchmark on state-of-the-art open-source and closed-source VLMs, analyzing their performance under various prompting strategies. The results confirm the significant promise of this code-based direction. Furthermore, we analyze how this code-based reconstruction paradigm benefits subsequent editing operations, and experimentally validated the superiority of this approach over traditional methods.

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

# A APPENDIX

## A.1 USE OF LLMs

We utilized Large Language Models (LLMs) to assist in preparing this paper in two primary ways:

- For polishing the language and phrasing of the text to enhance clarity, conciseness, and readability.
- For refining the prompts used to query the Vision Language Models (VLMs) in our experiments, to better align with effective prompt engineering principles.

## A.2 RECONSTRUCTION PIPELINE DETAILS

**ModelNet10 Easy/Hard Split.** We partition ModelNet10 into *easy* and *hard* subsets per category: objects with fewer parts, regular structures, and mild curvature transitions are labeled easy; objects with more parts, irregular topology, or pronounced/high-curvature transitions are labeled hard.

1. **bathtub**:
   *Easy*: bathtub_0111, bathtub_0119, bathtub_0139, bathtub_0154, bathtub_0155;
   *Hard*: bathtub_0141, bathtub_0153, bathtub_0124, bathtub_0115, bathtub_0150.

2. **Bed**:
   *Easy*: bed_0555, bed_0557, bed_0561, bed_0572, bed_0595;
   *Hard*: bed_0548, bed_0566, bed_0571, bed_0614, bed_0598.

3. **Chair**:
   *Easy*: chair_0894, chair_0897, chair_0950, chair_0901, chair_0896;
   *Hard*: chair_0893, chair_0891, chair_0941, chair_0898, chair_0943.

4. **Desk**:
   *Easy*: desk_0217, desk_0262, desk_0246, desk_0236, desk_0220;
   *Hard*: desk_0263, desk_0253, desk_0231, desk_0209, desk_0226.

5. **Dresser**:
   *Easy*: dresser_0248, dresser_0254, dresser_0266, dresser_0232, dresser_0205;
   *Hard*: dresser_0209, dresser_0257, dresser_0243, dresser_0217, dresser_0233.

6. **Monitor**:
   *Easy*: monitor_0503, monitor_0545, monitor_0535, monitor_0531, monitor_0528;
   *Hard*: monitor_0483, monitor_0522, monitor_0529, monitor_0511, monitor_0539.

7. **night_stand**:
   *Easy*: night_stand_0207, night_stand_0232, night_stand_0263, night_stand_0231, night_stand_0283;
   *Hard*: night_stand_0225, night_stand_0208, night_stand_0278, night_stand_0270, night_stand_0262.

8. **Sofa**:
   *Easy*: sofa_0692, sofa_0687, sofa_0770, sofa_0756, sofa_0683;
   *Hard*: sofa_0761, sofa_0777, sofa_0745, sofa_0746, sofa_0743.

9. **Table**:
   *Easy*: table_0443, table_0439, table_0399, table_0422, table_0447;
   *Hard*: table_0436, table_0405, table_0430, table_0470, table_0423.

10. **Toilet**:
    *Easy*: toilet_0393, toilet_0438, toilet_0408, toilet_0355, toilet_0439;
    *Hard*: toilet_0419, toilet_0409, toilet_0367, toilet_0436, toilet_0401.

**Failure rate of VLMs.** We report the proportion of prompts that produced *unsuccessful* runs on ModelNet10, i.e., the generated *bpy* script did not compile or crashed in headless Blender 4.4. For each task, when the first generated code runs into an error, the model has 5 chances to correct it. If the code still reports an error after the chances are exhausted, the generation is considered to be failed. The final failure rates are shown in Table 5 and Table 6. For open sourced models, we only

Table 5: **Open-source VLM failure rate** on ModelNet 10. Values are *fail/total*.

| Strategy | InternVL3.5-38B | Qwen2.5-VL-72B | LLaVA-OneVision-72B |
|---|---|---|---|
| Single-call | 4/100 (4%) | 4/100 (4%) | 48/100 (48%) |
| Blueprint | 24/100 (24%) | 2/100 (2%) | 40/100 (40%) |
| RAG | 22/100 (22%) | 6/100 (6%) | 45/100 (45%) |

Table 6: **Closed-source VLM failure rate** on ModelNet 10. Values are *fail/total*. In our experiment, no errors occur in o3 and Claude on "Single call" and "RAG" paradigm.

| Strategy | Gemini 2.5 Pro | Gemini 3.0 Pro |
|---|---|---|
| Single-call | 2/100 (2%) | 0/100 (0%) |
| Blueprint | 8/100 (8%) | 1/100 (1%) |
| RAG | 4/100 (4%) | 2/100 (2%) |

test metrics on correctly generated samples, while we fill in the failed samples of the closed source model before test the closed source model on the complete samples.

**Examples.** Blueprint example can be found at List 1, Blender api example at Figure 11, example of a query generated by Gemini-2.5-pro at Figure 12, retrieved RAG example at Figure 13.

A.3 MORE EXPERIMENTAL RESULTS

**Complete Results on ModelNet10-*easy* and *hard*.** We fully tested the three closed-source VLMs on both ModelNet10-*easy* and *hard* across the *Single-call*, *Planning*, and *RAG* pipelines; results are summarized in Table 7. Overall, the conclusions on the full setting are consistent with those reported in the main paper (where we focused on InstantMesh/Unique3D and the *RAG* variants of *o3* and *Gemini*). Code-based reconstruction generally outperforms hard parts on easy parts, while mesh-based reconstruction does not exhibit this phenomenon. This suggests that code-based reconstruction methods struggle to accurately restore complex structures, as VLMs struggle to accurately capture them.

**Reconstruction Bad Cases Analysis.** Figure 7 demonstrates several failure cases of Gemini-2.5-pro within the RAG paradigm. For Chair 0898, while the generated object exhibits a plausible shape and successfully produces chair legs with complex intersecting lines, it does not conform to the specifications of the ground truth. The reconstruction of Chair 0941 captures the general structure; however, the size of the "Y"-shaped backrest is incorrect, and the interconnecting components between the legs are missing. In Chair 0950, the individual components are generated approximately correctly, but their spatial arrangement is inaccurate, resulting in an overall structure that deviates significantly from the ground truth. At first glance, the object Table 0423 appears somewhat similar, but a detailed inspection reveals that the orientation of the legs and the angles of the connecting bars are rotated by 90 degrees. Furthermore, while the ground truth features four A-shaped leg structures, the generated object exhibits only two. For Desk 0217, the model misjudges the relative spacing, placing a horizontal bar at the midpoint instead of the correct position at one-quarter of the height. Desk 0226 possesses a complex structure with numerous curved elements and components. Although the final generated result bears a rough resemblance, the details differ substantially.

The primary failure modes can be summarized as follows:

- Incomplete comprehension of the input image, leading to missing components.
- Difficulty in accurately interpreting complex images, resulting in structures that are only coarsely similar to the ground truth.
- Insufficient spatial reasoning capability, causing failures in the correct assembly of components even when they are generated accurately.

**Depth and Normal rendering examples.** We render 3D objects from multiple perspectives, as shown in Figure 6. This ensures that the input image contains as much structural information as

Table 7: **Qualitative reconstruction results on ModelNet10 *easy/hard* split.** "Sin." stands for "single-call" and "Pla." for "planning". The best value in each block is highlighted in green, and the second best value is blue.

| Model | Paradigm | ModelNet10–*easy* | | | ModelNet10–*hard* | | |
|---|---|---|---|---|---|---|---|
| | | CD↓ | IoU↑ | F@5%↑ | CD↓ | IoU↑ | F@5%↑ |
| *Traditional baselines* | | | | | | | |
| Unique3D | — | 0.0557 | 0.1395 | 0.6151 | 0.0515 | 0.1544 | 0.6472 |
| InstantMesh | — | **0.0217** | **0.3153** | **0.8788** | **0.0219** | **0.2946** | **0.8831** |
| *Closed-source VLM families* | | | | | | | |
| | Sin. | 0.0327 | 0.2348 | 0.7845 | 0.0368 | 0.2191 | 0.7483 |
| Claude Sonnet 4.0 | Pla. | 0.0343 | 0.2522 | 0.7588 | 0.0383 | 0.2106 | 0.7481 |
| | RAG | 0.0338 | 0.2586 | 0.7784 | 0.0352 | 0.2124 | 0.7661 |
| | Sin. | 0.0437 | 0.1973 | 0.6941 | 0.0431 | 0.1702 | 0.7073 |
| o3 | Pla. | 0.0298 | 0.2481 | 0.8271 | 0.0359 | 0.2137 | 0.7638 |
| | RAG | 0.0281 | 0.2881 | 0.8208 | 0.0323 | 0.2246 | 0.8006 |
| | Sin. | 0.0385 | 0.2171 | 0.7220 | 0.0391 | 0.2103 | 0.7319 |
| Gemini 2.5 Pro | Pla. | 0.0277 | 0.2842 | 0.8419 | 0.0294 | 0.2552 | 0.8154 |
| | RAG | **0.0246** | **0.2977** | **0.8626** | **0.0285** | **0.2699** | **0.8288** |

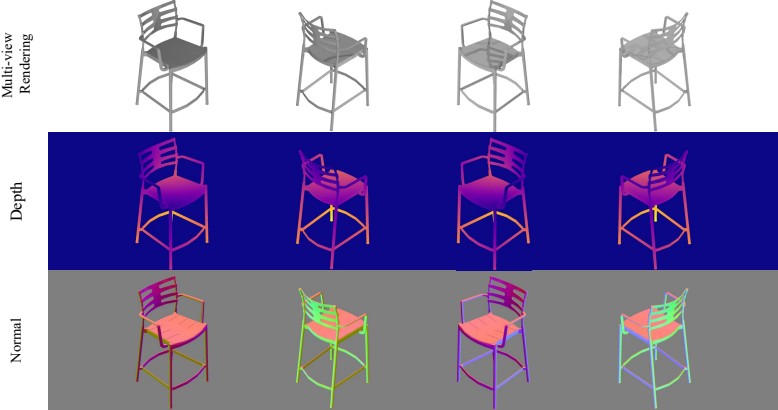

Figure 6: Rendered multi-view images of 3D object Chair 0891 in ModelNet10, with depth and surface normals.

possible. Furthermore, when testing 2D metrics, we can perform tests on images from multiple perspectives and take the average for a comprehensive evaluation.

**Tokens and time cost.** To better understand the practical overhead of different interaction paradigms, we report the average token usage and wall-clock time per call for Gemini 3.0 Pro in Table 8. The single-call paradigm is the most economical. In contrast, the planning paradigm roughly doubles the token footprint while the RAG paradigm is the most expensive. These results highlight a clear trade-off between efficiency and performance. In practice, this suggests that single-call prompting may be preferable in resource- or latency-sensitive scenarios, whereas planning/RAG are more suitable when accuracy is prioritized and moderate overhead is acceptable.

**Additional Qualitative Results.** We provide additional qualitative examples for both the **text-conditional reconstruction variant** in Figure 8 and for **code editing** in Figure 10. The specific text instructions used for each example of code editing are detailed in Section A.5.

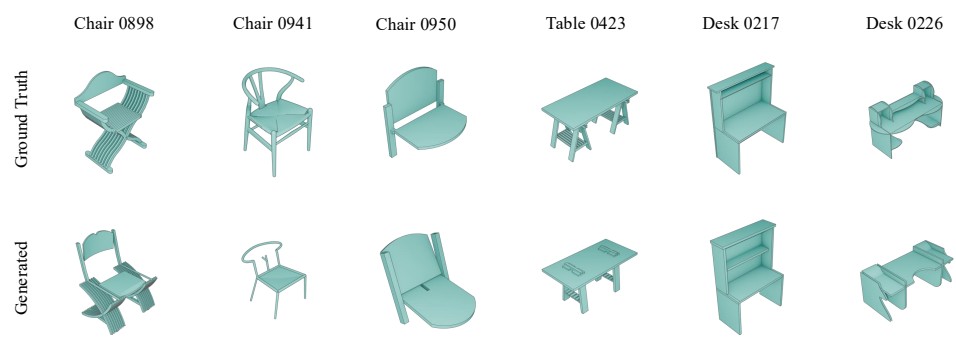

Figure 7: Failure cases demonstration. It can be seen that the semantic meaning of the object category is correct, but there may be deviations in the details.

Table 8: Average token usage and time per item for Gemini 3.0 Pro under different paradigms.

|  | single-call | planning | RAG |
|---|---|---|---|
| Prompt tokens | 1,222 | 9,093 | 11,867 |
| Completion tokens | 10,885 | 16,989 | 21,752 |
| Total tokens | 12,107 | 26,082 | 33,619 |
| Time per call (mins) | 2.27 | 3.17 | 4.00 |

**Articulation Results.** We evaluate the effectiveness of our Blender-based articulation pipeline on samples from three object categories: Cabinet, Monitor, and Toilet. Specifically, we utilize shape keys to implement the translational motion of cabinet drawers, while applying rotational transformations to achieve the horizontal and vertical swiveling of monitor screens and the axial rotation of toilet lids. All generated motions strictly adhere to realistic physical scenarios and kinematics. The qualitative results are illustrated in Figure 9.

## A.4 REFINEMENT PARADIGM DETAILS

**Refinement method.** Program-level refinement is an attractive direction for improving code-based 3D reconstruction. To this end, we implemented a refinement variant on top of our baseline.

We render the mesh generated by the RAG pipeline from three strategic viewpoints (`front_top`, `back_top`, `back_bottom`) to capture geometry that might be self-occluded. These renderings are fed into the VLM alongside the reference image $\mathcal{I}$ and the current script. The VLM is instructed to identify the most salient geometric mismatch (e.g., incorrect proportions or missing sub-components) and modify the code to rectify it. This process iterates until the model deems the reconstruction consistent with the reference or a maximum iteration limit ($N_{\text{refine}}$) is reached.

**Refinement result analysis.** As shown in Table 9, contrary to normal expectations, the performance of the model generally decreases after adding refinement. Under our current setup and existing VLM capability, a simple refinement pipeline cannot bring the expected benefits.

We believe this negative result is not contradictory to LL3M (Lu et al., 2025). LL3M is designed for text-to-3D asset creation with a multi-agent architecture. Its refinement loop is primarily evaluated qualitatively in terms of prompt faithfulness and user-controllable editing, not as a strict improvement over a known geometric ground truth. It does not guarantee that a single off-the-shelf VLM can reliably refine a reconstruction toward a fixed 3D ground truth. In contrast, our task is metric-driven single-image reconstruction against a fixed 3D target, where even small geometric mistakes are penalized. Our experiments indicate that naïve refinement in this regime is currently unreliable.

Our results are also consistent with the current limitations of off-the-shelf VLMs in precise 3D code editing. BlenderGym (Gu et al., 2025) inputs two images and an initial script, requesting VLM to

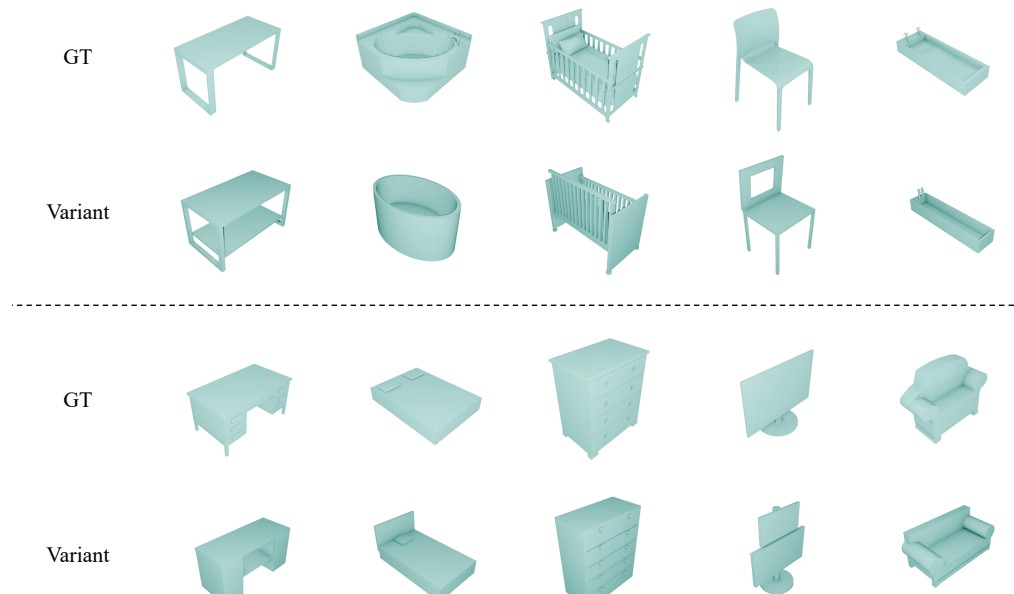

Figure 8: **Additional examples of the text-conditional reconstruction variant.** The model generates a modified 3D asset based on a source image from ModelNet10 and a corresponding text instruction.

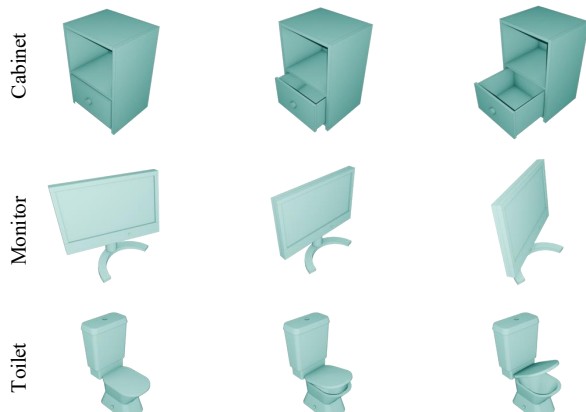

Figure 9: **Visualization of articulated objects.** We demonstrate the articulation effects generated by our Blender script across three categories. These include the sliding translation of cabinet drawers, the multi-axis rotation of monitor screens, and the hinge-based rotation of toilet lids, all simulating real-world usage scenarios.

modify the script to fit the target scene. BlenderGym also reports that large models such as GPT-4o may not identify minor visual differences, may have made mistakes, or have made changes unrelated to the differences. To make refinement somewhat useful, they need to use a generator–verifier multi-agent framework, multiple rounds of search/verification, and substantial computation, yet the results still fall far short of human performance. Moreover, our setting is more challenging than BlenderGym (reconstructing an entire object from scratch rather than editing an existing scene), so the instability of refinement is reasonable. Besides, IR3D-Bench (Liu et al., 2025) observes that current VLMs "grasp high-level object attributes" but "struggle with precise spatial control". Iterative refinement and careful prompt design can help, but require a dedicated agentic setup rather than a single-call VLM. In our iterative refinement experiment, we deliberately maintaining the zero-

Table 9: **Reconstruction on ModelNet10.** 3D and 2D metrics are the same as that in the main text. $N_{\text{refine}}$ equals three in our experiments. "Ref." stands for "Refinement".

| Model | Paradigm | ModelNet10 | | | | | |
| | | 3D Metrics | | | 2D Metrics | | |
| | | CD$\downarrow$ | 3D IoU$\uparrow$ | F@5%$\uparrow$ | NRMSE$\downarrow$ | SSIM$\uparrow$ | MAE$\downarrow$ |
| *Open-source VLM families* | | | | | | | |
| InternVL3.5-38B (Wang et al., 2025) | RAG | 0.0541 | 0.1675 | 0.6280 | 0.1198 | 0.8542 | 0.2062 |
| | Ref. | 0.0565 | 0.1614 | 0.6021 | 0.1532 | 0.8424 | 0.2202 |
| Qwen2.5-VL-72B-Instruct (Bai et al., 2025) | RAG | 0.0472 | 0.2009 | 0.6821 | 0.1071 | 0.8507 | 0.2042 |
| | Ref. | 0.0771 | 0.1143 | 0.4720 | 0.2102 | 0.7890 | 0.2514 |
| *Closed-source VLM families* | | | | | | | |
| Claude Sonnet 4.0 (Anthropic PBC, 2025) | RAG | 0.0345 | 0.2355 | 0.7723 | 0.0954 | 0.8913 | 0.1841 |
| | Ref. | 0.0352 | 0.2282 | 0.7642 | 0.0980 | 0.8884 | 0.1836 |
| o3 (OpenAI, 2025) | RAG | 0.0302 | 0.2564 | 0.8107 | 0.0830 | 0.9012 | 0.1635 |
| | Ref. | 0.0317 | 0.2517 | 0.8009 | 0.0899 | 0.8973 | 0.1592 |
| Gemini 2.5 Pro (Google DeepMind, 2025a) | RAG | 0.0266 | 0.2977 | 0.8626 | 0.0742 | 0.9093 | 0.1530 |
| | Ref. | 0.0276 | 0.2788 | 0.8388 | 0.0778 | 0.9077 | 0.1452 |
| Gemini 3.0 Pro (Google DeepMind, 2025b) | RAG | 0.0245 | 0.3024 | 0.8746 | 0.0770 | 0.9151 | 0.1430 |
| | Ref. | 0.0277 | 0.2743 | 0.8371 | 0.0836 | 0.9061 | 0.1517 |

shot and single-model setting. Without specialized agents or fine-tuning, it is difficult to consistently improve metrics, which aligns with the conclusions presented by IR3D-Bench.

From this perspective, our refinement experiments provide a useful empirical finding: simple one-shot or single-agent refinement with current off-the-shelf VLMs is not a reliable way to improve code-based 3D reconstruction, and can even be detrimental. Designing a full-fledged multi-agent refinement system is a highly promising direction, but orthogonal to the main contribution of our paper (a unified benchmark + analysis of VLM-based Blender-code reconstruction). We therefore view this as important future work.

## A.5 EDITING PIPELINE DETAILS

**Complete Instructions Used for Editing.** For the sake of aesthetics, we only show the most critical text instruction when showing the edit figures, and omit the long part with . . . . Here we list the complete instructions.

In Figure 1, the instructions we use in the right part are:

1. Open the lid of this cup.
2. The legs of this table are out of proportion to the tabletop. Optimize it by making the legs more slender.

In Figure 4, the instructions we use are:

1. Make the bathtub more square and add a flat base for stability.
2. Add a second drawer below the existing one.
3. Change the base legs to a single centered pedestal.
4. Replace the cylindrical lampshade above this desk lamp with a triangular cone.
5. Make this table lamp taller. The column mistakenly passes through the lampshade and protrudes a little from the top. Remove this small part.
6. Lengthen the four cylindrical legs of this table and connect X-shaped wooden strips at the bottom of the four legs that is connect the legs at opposite corners at the bottom to make its structure more stable.

In Figure 5, the instructions we use are:

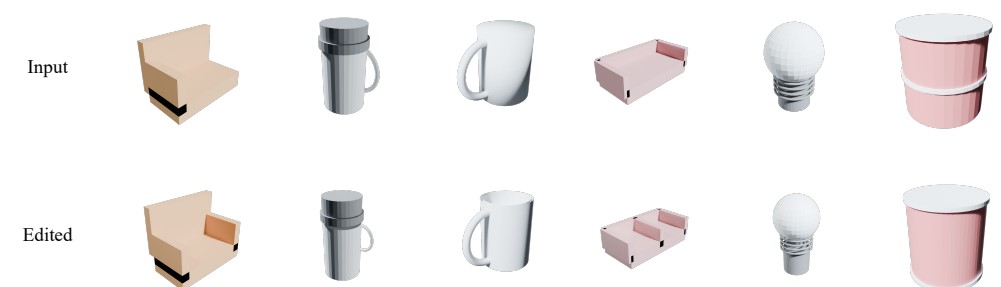

Figure 10: **Further examples of code editing on the BlendNet-E dataset.** These results show targeted geometric modifications based on textual instructions.

- Left part:
  1. Let all the rings on the pillar sink with gravity and fit together.
  2. The ring handle on the side of this cup is too big and does not match the cup body. Make it smaller.
  3. The candle on this cake is too thick and short. Make it thinner and longer and reduce the number to 1 and insert it in the middle of the top of the cake.
- Right part:
  1. Let all the rings on the pillar sink with gravity and fit together.
  2. The ring handle on the side of this cup is too big and does not match the cup body. Make it smaller.
  3. The candle on this cake is too thick and short. Make it thinner and longer and reduce the number to 1 and insert it in the middle of the top of the cake.

In Figure 8, the instructions we use are:

- Upper part:
  1. Add a lower shelf between the two legs.
  2. Convert the corner bath to an oval shape.
  3. Convert one of the crib's sides into a removable panel.
  4. Cut a large opening in the middle of the backrest.
  5. Extend the basin to double its current length.
- Lower part:
  1. Add a central open shelf in the knee space area for additional storage.
  2. Add a headboard to the bed.
  3. Add a fifth drawer at the bottom.
  4. Add a second, smaller screen on top to create a dual-monitor setup.
  5. Add a lower central support beam between the sofa legs.

In Figure 10, the instructions we use are:

1. This sofa has armrests on only one side and the modification makes it have armrests on both sides.
2. The keychain circle on this cup is too big, make it smaller
3. The cylindrical portion of this cup was incorrectly generated as a solid shape. Make it hollow.
4. Add a handguard in the middle of this sofa to give it two separate seats.
5. Separate the spherical part of this bulb from the base.
6. This bucket has an ugly ring around the cylindrical waist. Remove it.

## A.6 DETAILED PIPELINE ARTIFACTS

Listing 1: Blueprint JSON for object in Figure 2. The blueprint normalizes to a base dimension (`overall_height`=1.0) and encodes part-wise parametrics used by the code generator.

```json
1  {
2    "object_category": "bar_stool",
3    "base_dimension": "overall_height",
4    "dimensional_profile": {
5      "overall_height": 1.0,
6      "overall_width_at_base_ratio_to_height": 0.42,
7      "overall_depth_at_base_ratio_to_height": 0.4,
8      "seat_height_from_ground_ratio_to_overall_height": 0.65
9    },
10   "geometric_components": {
11     "legs": {
12       "count": 4,
13       "profile_shape": "cylindrical",
14       "diameter_ratio_to_overall_height": 0.02,
15       "splay_angle_from_vertical_degrees": 6.0
16     },
17     "footrest": {
18       "structure_type": "continuous_four_sided_brace",
19       "height_from_ground_ratio_to_overall_height": 0.18,
20       "cross_section_diameter_ratio_to_leg_diameter": 1.0,
21       "front_bar_outward_curve_depth_ratio_to_overall_depth": 0.15
22     },
23     "seat": {
24       "plan_shape": "rounded_square",
25       "width_ratio_to_overall_width_at_base": 0.86,
26       "depth_ratio_to_overall_depth_at_base": 0.85,
27       "thickness_ratio_to_overall_height": 0.03,
28       "ergonomic_concave_dip_ratio_to_seat_depth": 0.05,
29       "front_edge_waterfall_radius_ratio_to_seat_thickness": 1.0,
30       "cutouts": {
31         "count": 2,
32         "type": "slot",
33         "slot_length_ratio_to_seat_depth": 0.7,
34         "slot_width_ratio_to_seat_width": 0.08,
35         "slot_spacing_from_centerline_ratio_to_seat_width": 0.25
36       }
37     },
38     "backrest": {
39       "height_above_seat_ratio_to_overall_height": 0.35,
40       "width_ratio_to_seat_width": 0.95,
41       "tilt_angle_from_vertical_degrees": 12.0,
42       "horizontal_lumbar_curve_depth_ratio_to_width": 0.08,
43       "structure": {
44         "type": "slatted_frame",
45         "frame_thickness_ratio_to_leg_diameter": 1.2,
46         "slat_count": 3,
47         "slat_height_ratio_to_backrest_height": 0.12,
48         "slat_vertical_gap_ratio_to_slat_height": 1.1,
49         "vertical_support_count": 2,
50         "vertical_support_width_ratio_to_frame_thickness": 1.0
51       }
52     },
53     "armrests": {
54       "count": 2,
55       "structure_type": "continuous_loop_from_backrest_to_seat",
56       "height_above_seat_at_rear_ratio_to_overall_height": 0.15,
57       "length_ratio_to_seat_depth": 0.8,
58       "cross_section_diameter_ratio_to_leg_diameter": 1.0,
59       "downward_slope_angle_degrees": 3.0,
60       "outward_bow_distance_ratio_to_seat_width": 0.05
61     }
62   }
63 }
```

```json
{
  "symbol": "bpy.ops.curves.add_bezier",
  "language": "python",
  "module": "bpy.ops.curves",
  "signature": ".ops.curves.add_bezier(*, radius=1.0, enter_editmode=False,
      align='WORLD', location=(0.0, 0.0, 0.0), rotation=(0.0, 0.0, 0.0), scale=(0.0,
      0.0, 0.0))",
  "parameters": [
    {"name": "radius", "description": "radius (float in [0, inf], (optional))
        Radius"},
    {"name": "enter_editmode", "description": "enter_editmode (boolean, (optional))
        Enter Edit Mode, Enter edit mode when adding this object"},
    {"name": "align", "description": "align (enum in ['WORLD', 'VIEW', 'CURSOR'],
        (optional))    Align, The alignment of the new object\n\nWORLD\nWorld
        Align the new object to the world.\nVIEW\nView    Align the new object to
        the view.\nCURSOR\n3D Cursor    Use the 3D cursor orientation for the new
        object.\n\n"},
    {"name": "location", "description": "location (mathutils.Vector of 3 items in [-inf,
        inf], (optional))    Location, Location for the newly added object"},
    {"name": "rotation", "description": "rotation (mathutils.Euler rotation of 3 items
        in [-inf, inf], (optional))    Rotation, Rotation for the newly added
        object"},
    {"name": "scale", "description": "scale (mathutils.Vector of 3 items in [-inf, inf],
        (optional))    Scale, Scale for the newly added object"}
  ],
  "doc_text": "Add new bezier curve",
  "version": "4.4"
}
```

Figure 11: Blender 4.4 Python API database entry (structured JSON).

```
1  {
2    "intent_queries": {
3      "queries": {
4        "legs": {
5          "prefer_modules": ["bpy.ops.mesh", "bpy.ops.object"],
6          "keywords": ["primitive_cylinder_add", "cylinder", "duplicate_move", "rotate",
             ↪ "splay"],
7          "query": "Create four cylindrical legs for a bar stool. Start by adding a
               ↪ cylinder primitive, then duplicate it and rotate the legs to create a
               ↪ splayed angle from the vertical."
8        },
9        "footrest": {
10         "prefer_modules": ["bpy.ops.curve", "bpy.ops.object"],
11         "keywords": ["curve", "bezier", "bevel", "extrude", "depth", "join"],
12         "query": "Model a continuous four-sided footrest brace with a curved front. Use
               ↪ a bezier curve with bevel depth to form the cylindrical rails and join
               ↪ the segments."
13       },
14       "seat": {
15         "prefer_modules": ["bpy.ops.mesh", "bpy.ops.object"],
16         "keywords": ["primitive_cube_add", "subdivision_set", "bevel", "boolean",
             ↪ "proportional_edit", "loop_cut"],
17         "query": "Create a rounded square seat with an ergonomic dip and two slot
               ↪ cutouts. Start with a cube, use subdivision and proportional editing for
               ↪ the dip, bevel the edges, and apply a boolean difference modifier for the
               ↪ slots."
18       },
19       "backrest": {
20         "prefer_modules": ["bpy.ops.mesh", "bpy.ops.object"],
21         "keywords": ["plane", "extrude", "inset", "boolean", "loop_cut", "curve_deform",
             ↪ "modifier"],
22         "query": "Create a slatted backrest frame that is tilted and curved. Model the
               ↪ basic shape from a plane using extrude and inset, use a boolean modifier
               ↪ to create the slats, and bend the result with a curve deform modifier."
23       },
24       "armrests": {
25         "prefer_modules": ["bpy.ops.curve", "bpy.ops.object"],
26         "keywords": ["curve", "bezier", "extrude", "bevel", "depth", "mirror"],
27         "query": "Create two continuous loop armrests extending from the backrest to the
               ↪ seat. Model one armrest using a bezier curve with a circular bevel depth,
               ↪ then use a mirror modifier to create the second one."
28       }
29     }
30   }
31 }
```

Figure 12: VLM-generated intent queries per blueprint component.

```
1   {
2     "components": [
3       {
4         "name": "legs",
5         "apis": [
6           {
7             "symbol": "bpy.ops.mesh.primitive_cylinder_add",
8             "signature": "primitive_cylinder_add(vertices=32, radius=1.0, depth=2.0,
                  ↪ end_fill_type='NGON', calc_uvs=True, enter_editmode=False,
                  ↪ align='WORLD', location=(0.0, 0.0, 0.0), rotation=(0.0, 0.0, 0.0),
                  ↪ scale=(0.0, 0.0, 0.0))",
9             "parameters": {
10              "vertices": "int in [3, 10000000], optional – The number of vertices for the
                    ↪ cylinder's circular caps.",
11              "radius": "float in [0, inf], optional – The radius of the cylinder.",
12              "depth": "float in [0, inf], optional – The depth (height) of the cylinder.",
13              "end_fill_type": "enum in ['NOTHING', 'NGON', 'TRIFAN'], optional – The
                    ↪ method to fill the ends of the cylinder.",
14              "location": "3D vector, optional – Location for the newly added object.",
15              "rotation": "3D Euler rotation, optional – Rotation for the newly added
                    ↪ object.",
16              "scale": "3D vector, optional – Scale for the newly added object."
17            }
18          }
19        ]
20      },
21      { "name": "footrest", "apis": [] },
22      { "name": "seat", "apis": [] },
23      { "name": "backrest", "apis": [] },
24      {
25        "name": "armrests",
26        "apis": [
27          {
28            "symbol": "bpy.ops.curves.add_bezier",
29            "signature": "add_bezier(radius=1.0, enter_editmode=False, align='WORLD',
                  ↪ location=(0.0, 0.0, 0.0), rotation=(0.0, 0.0, 0.0), scale=(0.0, 0.0,
                  ↪ 0.0))",
30            "parameters": {
31              "radius": "float in [0, inf], optional – The radius to set for the curve
                    ↪ points.",
32              "enter_editmode": "boolean, optional – If true, enter Edit Mode after
                    ↪ creating the object.",
33              "align": "enum in ['WORLD', 'VIEW', 'CURSOR'], optional – The alignment of
                    ↪ the new object. WORLD: Align the new object to the world. VIEW: Align
                    ↪ the new object to the view. CURSOR: Use the 3D cursor orientation for
                    ↪ the new object."
34            }
35          }
36        ]
37      }
38    ]
39  }
```

Figure 13: Retrieved APIs organized by component (post-retrieval structuring).

