# OpenReview forum: "3D-CoS: A New 3D Reconstruction Paradigm Based on VLM Code Synthesis"
_ICLR.cc/2026/Conference — Submitted to ICLR 2026_

### Official Review · Reviewer_n5AR · 2025-10-26

**Soundness:** 2
**Presentation:** 3
**Contribution:** 2
**Rating:** 4
**Confidence:** 4

**Summary:**

This work introduces 3D-Cos, a reconstruction framework that leverages vision-language models (VLMs) to generate 3D models as Blender Python code, given single images as input. To improve reconstruction quality, the authors propose a planning stage for blueprint inference and Retrieval-Augmented Generation (RAG) over Blender API documentation. A code-based reconstruction benchmark is introduced to evaluate the capabilities of existing VLMs in 3D reconstruction and 3D editing tasks. Results show that planning and RAG improve 3D-Cos's reconstruction performance, and that code-based 3D representation has advantages in 3D editing.

**Strengths:**

- This work treats 3D reconstruction as a Python code generation problem, which allows fine-grained control and editing of the generated 3D models.
- The proposed planning stage and RAG over Blender API documentation make sense and are shown to improve reconstruction quality over the single-call paradigm.
- The authors introduce a comprehensive benchmark for Blender code reconstruction from images, and they evaluate several existing VLMs, including both open-source and closed-source models, on this benchmark.
- The paper is well-written and easy to follow.

**Weaknesses:**

- I would view this work as more of an evaluation paper than a methodological paper, as the proposed techniques (planning and RAG) are mostly prompt engineering efforts. While the idea of reconstructing 3D models via code generation is interesting and useful, the technical novelty is limited.
- In the proposed pipeline, the authors consider only planning and RAG, but other techniques such as in-context learning or iterative refinement were not explored, making the study less comprehensive.
- The proposed approach is limited to 3D shapes with relatively simple geometries. It would struggle to handle more complex shapes, such as organic shapes, as discussed in the limitations section.

**Questions:**

- As mentioned in the weaknesses, planning and RAG may not be sufficient to ensure 3D reconstruction quality. It might be helpful to leverage the in-context learning capabilities of VLMs and prompt the models with a few examples. For challenging shape reconstruction, a single pass of the proposed pipeline may not guarantee satisfactory results. Incorporating a critic model to evaluate the generated shapes and provide feedback for iterative refinement could be beneficial, as demonstrated in a concurrent work LL3M [Lu et al. 2025].
- The proposed method focuses on single-image reconstruction. This is a challenging task of course, but it would be interesting to see how the method performs with multi-view inputs, which should lead to better reconstruction quality.
- The proposed benchmark seems not very comprehensive, for example, it lacks ground-truth for 3D editing tasks. While there is CLIP score evaluation, a perceptual user study to evaluate the quality of the reconstructed shapes is missing.
- I did not find a promise of code and data release, which would limit the reproducibility.

Minor:
- When using "3D edition" in the several places of the main text, I guess the authors mean "3D editing".
- Fig. 7's caption is missing.

---

> ### Author Response · Authors · 2025-11-26
> **Response to Reviewer n5AR**
>
> **Q1: More of an evaluation paper than a methodological paper.**
>
> A1: We thank the reviewer for pointing this out. However, we believe the paper makes concrete methodological contributions beyond being purely exploratory. As shown in Table 1, even when using the strongest model (e.g., Gemini 3.0 Pro), the naive Single-call strategy performs clearly worse than our designed few-shot code reconstruction paradigm.
>
> While planning, RAG, and in-context learning are known ideas, our work integrates those ideas into a unified few-shot scheme tailored for code-based 3D reconstruction. The few-shot exemplars jointly teach (i) how to produce quantitative blueprints and (ii) how to invoke Blender APIs correctly. As a result, this concrete realization allows our code-based pipeline to reach, and sometimes surpass the leading supervised mesh baseline *InstantMesh*.
>
> We also introduce and analyze a new methodological paradigm for 3D reconstruction and editing: representing 3D assets as executable Blender code generated by VLMs (3D-CoS), and to use extensive experiments to quantify the current capability frontier of VLMs under this paradigm. The pipeline, datasets, and evaluation protocol together form a reusable methodological framework for future work, beyond a one-off exploratory case study.
>
> **Q2: In-context learning and iterative refinement were not explored.**
>
> A2.1 (in-context learning) We appreciate the valuable comment. To investigate the in-context learning capabilities of VLMs, we prompt the model with a concatenated input consisting of a few exemplary pairs. Each exemplar includes (i) a rendered image of an object and (ii) a carefully curated Blender script that showcases correct usage of our quantitative blueprint logic and appropriate Blender API calls. To prevent data leakage, these examples are selected distinct from our test set ModelNet10. Under this few-shot setting, VLMs generally improve over their RAG-only counterparts. A subset of the results is shown below (full results are reported in Table 1 of the current paper draft):
>
> | Model             | Paradigm | CD ↓     | 3D IoU ↑   | F@5% ↑   | NRMSE ↓   | SSIM ↑   | MAE ↓   |
> |-------------------|----------|----------|------------|----------|-----------|----------|---------|
> | Claude Sonnet 4.0 | RAG      | 0.0345   | 0.2355     | 0.7723   | 0.0954    | **0.8913** | **0.1841** |
> |                   | Few-shot     | **0.0333** | **0.2398** | **0.7833** | **0.0927** | 0.8893   | 0.2217 |
> | o3                | RAG      | 0.0302   | 0.2564     | 0.8107   | **0.0830** | 0.9012   | **0.1635** |
> |                   | Few-shot     | **0.0283** | **0.2781** | **0.8204** | 0.0834    | **0.9021** | 0.1789 |
> | Gemini 3.0 Pro    | RAG      | 0.0245   | 0.3024     | 0.8746   | 0.0770    | 0.9151   | **0.1430** |
> |                   | Few-shot     | **0.0223** | **0.3083** | **0.8824** | **0.0659** | **0.9212** | 0.1492 |
>
> (Continued on next comment)

---

> > ### Author Response · Authors · 2025-11-26
> > **Response to Reviewer n5AR (cont'd)**
> >
> > A2.2 (iterative refinement) We agree that program-level refinement is an attractive direction for improving code-based 3D reconstruction, and we thank the reviewer for pointing to LL3M. In response to this comment, we have conducted additional experiments implementing a visual refinement pipeline. However, contrary to normal expectations, under our current setup（no specialized agents or fine-tuning）and existing VLM capability, a simple refinement pipeline can not bring the expected benefits.
> >
> > | Model              | Method | CD ↓     | 3D IoU ↑   | F@5% ↑   | NRMSE ↓   | SSIM ↑   | MAE ↓   |
> > |--------------------|--------|----------|------------|----------|-----------|----------|---------|
> > | Claude Sonnet 4.0  | RAG    | **0.0345** | **0.2355** | **0.7723** | **0.0954** | **0.8913** | 0.1841 |
> > |                    | Refinement   | 0.0352   | 0.2282     | 0.7642   | 0.0980    | 0.8884   | **0.1836** |
> > | o3                 | RAG    | **0.0302** | **0.2564** | **0.8107** | **0.0830** | **0.9012** | 0.1635 |
> > |                    | Refinement   | 0.0317   | 0.2517     | 0.8009   | 0.0899    | 0.8973   | **0.1592** |
> > | Gemini 3.0 Pro     | RAG    | **0.0245** | **0.3024** | **0.8746** | **0.0770** | **0.9151** | **0.1430** |
> > |                    | Refinement   | 0.0277   | 0.2743     | 0.8371   | 0.0836    | 0.9061   | 0.1517 |
> >
> > - We believe this negative result is not contradictory to LL3M. LL3M is designed for text-to-3D asset creation, evaluated qualitatively in terms of prompt faithfulness. It does not guarantee that a single off-the-shelf VLM can reliably refine a reconstruction toward a fixed 3D ground truth. In contrast, our task is single-image reconstruction against a fixed 3D target, where even small geometric mistakes are penalized.
> > - Our results are also consistent with the current limitations of off-the-shelf VLMs in precise 3D code editing. BlenderGym also reports that large models such as GPT-4o may not identify minor visual differences, or have made changes unrelated to the differences. To make refinement somewhat useful, they need to use a generator–verifier multi-agent framework, multiple rounds of search/verification, and substantial computation, yet the results still fall far short of human performance. Moreover, our setting is more challenging than BlenderGym (reconstructing an entire object from scratch rather than editing an existing scene), so the instability of refinement is reasonable.
> > - details of these results can be found in Appendix A.4
> >
> > **Q3: The proposed method focuses on single-image reconstruction.**
> >
> > A3: We thank the reviewer for pointing this out, and we have conducted preliminary multi-image experiments showing clear gains. However, our current experiments intentionally focus on single-image inputs. Our primary goal is to stress-test whether VLMs within the 3D-CoS framework can produce plausible and geometrically consistent completions when the visible field of view is partial or occluded, which is also why we designed the 2D view-based metrics.
> >
> > | Model              | Method | CD ↓     | 3D IoU ↑   | F@5% ↑   |
> > |--------------------|--------|----------|------------|----------|
> > | Gemini 2.5 Pro     | Single-image   | 0.0388   | 0.2137     | 0.7269   |
> > |                    | Multi-images   | **0.0351** | **0.2505** | **0.7660** |
> >
> > **Q4: User study to evaluate 3D editing quality.**
> >
> > A4: We thank the reviewer for this insightful suggestion. We already include a user study targeting 3D editing quality in the current draft (Table 4), assessing "Instruction following" and "Preservation of unedited regions". In both dimensions, our method shows significant advantages over BlendedPC.
> > | Method              | Instruction following  | Preservation of unedited regions   |
> > |--------------------|--------|----------|
> > | BlendedPC     |  1.90   | 2.45   |
> > | Ours                   |  **4.37** | **4.30** |
> >
> > **Q5: A promise of code and data release and other minor problems.**
> >
> > A5: We appreciate the valuable reminder and we apologize for this omission. Upon acceptance, we commit to releasing:
> > - the full 3D-CoS implementation (blueprint prediction, Blender-code generation, RAG over Blender docs)
> > - all evaluation code, including the metrics used for 3D and 2D evaluation
> > - the processed datasets and metadata used in our reconstruction experiments
> >
> > Besides, all the minor problems has been tackled in the current draft.

---

### Official Review · Reviewer_gLSo · 2025-11-01

**Soundness:** 2
**Presentation:** 1
**Contribution:** 2
**Rating:** 4
**Confidence:** 3

**Summary:**

TLDR: Systematic evaluation of off-the-shelf VLMs for image-to-Blender-code 3D reconstruction without fine-tuning.

This paper benchmarks VLMs (Gemini, o3, Claude, etc.) on generating Blender Python code from single RGB images. Evaluates three paradigms (Single-call, Planning, RAG) on ModelNet10. Best results lag behind InstantMesh, especially on complex shapes.

**Strengths:**

- Comprehensive benchmark of VLM capabilities (open-source + closed-source)
- Unified evaluation protocol with registration and multi-view metrics
- Demonstrates zero-shot capability (no fine-tuning) works reasonably for simple shapes
- RAG + Planning improvements are well-motivated

**Weaknesses:**

- No feedback loop: only error correction, no visual/quality-based refinement
- Questionable whether zero-shot approach is viable without 3D knowledge injection, basically rely on the base model.
- Poor presentation quality: inconsistent color coding in figures (teal/orange/colorful), confusing visualization
- Figure 7 has template caption "Caption" instead of actual description - indicates low attention to detail
- Missing citations of seminal works on shape programs/DSLs (e.g., Kenny Jones et al.'s ShapeAssembly and related foundational papers that pioneered code-based 3D representation)

**Questions:**

Why 3D shapes in Figures use different color code?
How do you compare with MeshCoder?

---

> ### Author Response · Authors · 2025-11-26
> **Response to Reviewer gLSo**
>
> **Q1: Concerns about "No feedback loop" and lack of visual/quality-based refinement.**
>
> A1: We agree that a visual feedback loop is an attractive direction for improving code-based 3D reconstruction, and we thank the reviewer for this insightful suggestion. In response to this comment, we have conducted additional experiments implementing a visual refinement pipeline. However, contrary to normal expectations, under our current setup and existing VLM capability, a simple refinement pipeline can not bring the expected benefits.
>
> | Model              | Method | CD ↓     | 3D IoU ↑   | F@5% ↑   | NRMSE ↓   | SSIM ↑   | MAE ↓   |
> |--------------------|--------|----------|------------|----------|-----------|----------|---------|
> | Claude Sonnet 4.0  | RAG    | **0.0345** | **0.2355** | **0.7723** | **0.0954** | **0.8913** | 0.1841 |
> |                    | Refinement   | 0.0352   | 0.2282     | 0.7642   | 0.0980    | 0.8884   | **0.1836** |
> | o3                 | RAG    | **0.0302** | **0.2564** | **0.8107** | **0.0830** | **0.9012** | 0.1635 |
> |                    | Refinement   | 0.0317   | 0.2517     | 0.8009   | 0.0899    | 0.8973   | **0.1592** |
> | Gemini 3.0 Pro     | RAG    | **0.0245** | **0.3024** | **0.8746** | **0.0770** | **0.9151** | **0.1430** |
> |                    | Refinement   | 0.0277   | 0.2743     | 0.8371   | 0.0836    | 0.9061   | 0.1517 |
>
> - We believe this negative result is not contradictory to LL3M. LL3M is designed for text-to-3D asset creation, evaluated qualitatively in terms of prompt faithfulness. It does not guarantee that a single off-the-shelf VLM can reliably refine a reconstruction toward a fixed 3D ground truth. In contrast, our task is single-image reconstruction against a fixed 3D target, where even small geometric mistakes are penalized.
> - Our results are also consistent with the current limitations of off-the-shelf VLMs in precise 3D code editing. BlenderGym also reports that large models such as GPT-4o may not identify minor visual differences, or have made changes unrelated to the differences. To make refinement somewhat useful, they need to use a generator–verifier multi-agent framework, multiple rounds of search/verification, and substantial computation, yet the results still fall far short of human performance. Moreover, our setting is more challenging than BlenderGym (reconstructing an entire object from scratch rather than editing an existing scene), so the instability of refinement is reasonable.
> - details of these results can be found in Appendix A.4
>
> **Q2: Questionable viability of zero-shot approach without fine-tuning.**
>
> A2.1 We thank the reviewer for raising this fundamental question regarding the viability of relying on base models. Our primary objective in this work is to delineate the capability frontier of off-the-shelf VLMs. We aim to rigorously test how well general-purpose models can reason about 3D space using their pre-trained internal knowledge, without the resource-intensive process of 3D-specific training.
>
> A2.2 To further investigate the potential of base models with lightweight knowledge injection, we introduced a Few-shot paradigm (In-context Learning) in the revised manuscript. Notably, Gemini 3.0 Pro achieves a 3D IoU of 0.3083 and F-score of 0.8824, which surpasses the leading supervised mesh baseline, \textit{InstantMesh} (0.3049 / 0.8809). This proves that the approach relying on base models is not only viable but highly competitive. A subset of the results is shown below (full results are reported in Table 1 of the current paper draft):
>
> | Model             | Paradigm | CD ↓     | 3D IoU ↑   | F@5% ↑   | NRMSE ↓   | SSIM ↑   | MAE ↓   |
> |-------------------|----------|----------|------------|----------|-----------|----------|---------|
> | Claude Sonnet 4.0 | RAG      | 0.0345   | 0.2355     | 0.7723   | 0.0954    | **0.8913** | **0.1841** |
> |                   | Few-shot     | **0.0333** | **0.2398** | **0.7833** | **0.0927** | 0.8893   | 0.2217 |
> | o3                | RAG      | 0.0302   | 0.2564     | 0.8107   | **0.0830** | 0.9012   | **0.1635** |
> |                   | Few-shot     | **0.0283** | **0.2781** | **0.8204** | 0.0834    | **0.9021** | 0.1789 |
> | Gemini 3.0 Pro    | RAG      | 0.0245   | 0.3024     | 0.8746   | 0.0770    | 0.9151   | **0.1430** |
> |                   | Few-shot     | **0.0223** | **0.3083** | **0.8824** | **0.0659** | **0.9212** | 0.1492 |
>
> A2.3 We fully agree that extensive 3D knowledge injection via fine-tuning would further boost performance. Our training-free approach already rivals supervised baselines, which suggests that a fine-tuned "3D-Code-VLM" could indeed revolutionize this field. We view our work as a foundation justifying this direction, and we have discussed this potential in the Conclusion and Limitations sections.
>
> (Continued on next comment)

---

> > ### Author Response · Authors · 2025-11-26
> > **Response to Reviewer gLSo (cont'd)**
> >
> > **Q3: Comparison with *MeshCoder*.**
> >
> > A3: We thank the reviewer for pointing out this valid alternative. While *MeshCoder*[1] is a strong baseline for converting 3D data into code, our Image $\to$ Code (3D-CoS) approach offers distinct advantages.
> >
> > *MeshCoder* relies on a "part-to-code" inference model trained on specific synthetic datasets. Thus, it struggles with shapes outside its training distribution. As noted in their Appendix (A.1.2), *MeshCoder* requires designing dedicated procedural functions for specific objects like "spoons" and "forks" to handle their geometry. In contrast, our 3D-CoS leverages the pre-trained, open-world knowledge of VLMs. It can reconstruct arbitrary categories by composing general-purpose Blender API calls without being restricted to a fixed set of learned primitives.
> >
> > Regarding the input format, *MeshCoder* strictly requires complete 3D point clouds as input. To use *MeshCoder* on an image, one would first need to run an "Image-to-3D" reconstruction, which often produces noisy or non-manifold meshes. Converting these imperfect meshes into code is an ill-posed problem that leads to error accumulation. Our method requires only a single RGB image, utilizing the VLM's internal 3D priors to infer geometry and structure directly, making it significantly more applicable to real-world scenarios where only 2D observations are available.
> >
> > In summary, while our work and *MeshCoder* operate with different input modalities and application contexts, we share a unified vision of advancing code as a fundamental representation for 3D objects.
> >
> > **Q4: Missing citations.**
> >
> > A4: We apologize for the oversight in the initial submission and thank the reviewer for pointing out these seminal works. In the revised manuscript, we have expanded the "Code Based 3D Representations" subsection to explicitly discuss these foundational contributions, including but not limited to ShapeAssembly (Jones et al., 2020) and ShapeMOD (Jones et al., 2021). For details, please refer to "related work". If we are still missing important references, we would greatly appreciate further suggestions.
> >
> > **Q5: Why 3D shapes in Figures use different color code?**
> >
> > A5: We thank the reviewer for raising this question regarding visualization. The color coding is intentional and designed to distinguish between different tasks and datasets for better clarity:
> >
> > - For Reconstruction (ModelNet10): Since the ModelNet10 dataset does not possess color information and our task focuses purely on geometry, we render the input images in Grey and the generated 3D models in Cyan (as seen in Figures 1 and 3). This high-contrast Cyan shading is chosen specifically to highlight geometric details and topological structures against the background.
> >
> > - For Editing (BlendNet-E): These objects possess color attributes (e.g., brown, white). As shown in Figures 5 and 10, we render them in their original colors. This demonstrates that our code-based editing modifies the geometry while correctly preserving the object's original visual identity and color appearance.
> >
> > - For Point Clouds: In specific cases like Figures 1 and 4, we use a multi-colored rendering to represent objects in a point cloud state, distinguishing this modality from mesh geometry.
> >
> > **Q6: Presentation quality.**
> >
> > A6: We sincerely apologize for the template caption in Figure 7 and the confusion regarding the color coding. We thank the reviewer for the careful reading and for pointing out these presentation issues. In the revised manuscript, we have corrected the caption in Figure 7 to accurately describe the failure cases. Additionally, we clarified that the distinct color schemes are intentional design choices to distinguish between reconstruction (emphasizing geometry) and editing (emphasizing identity), as detailed in A5 above.
> >
> > [1] Bingquan Dai and Li Ray Luo and Qihong Tang and Jie Wang and Xinyu Lian and Hao Xu and Minghan Qin and Xudong Xu and Bo Dai and Haoqian Wang and Zhaoyang Lyu and Jiangmiao Pang. MeshCoder: LLM-Powered Structured Mesh Code Generation from Point Clouds. arXiv:2508.14879

---

### Official Review · Reviewer_WYcf · 2025-11-01

**Soundness:** 3
**Presentation:** 2
**Contribution:** 2
**Rating:** 6
**Confidence:** 3

**Summary:**

3D-CoS foucs on 3D reconstruction with Code Synthesis, where 3D objects as executable Blender Python code produced by large vision-language models (VLMs).

The authors propose a unified benchmark to systematically evaluate both open- and closed-source VLMs in this setting, focusing on single-image reconstruction and code-based editing. They also introduce two key improvements: a planning stage that produces part-level blueprints before code generation, and a retrieval-augmented generation (RAG) step leveraging Blender API documentation. Expeirment shows these two method enhenace the reconsturciton.

**Strengths:**

- The paper presents a valuable benchmarking effort for evaluating VLMs on code-based 3D reconstruction.
- The proposed technique is planning and rag shown to work effectively.
- Overall, the benchmarking design and experimental validation represent a meaningful step toward understanding the potential of code-based 3D modeling.

**Weaknesses:**

This is a compelling paradigm, but its practical scope is likely to be highly constrained.

- While the idea is interesting, the paper reads more as an insight-driven study rather than a technical contribution.
- Many of the observed improvements may naturally result from future advances in general-purpose VLM/LLM rather than from the specific domain techniques proposed here.

- As such, the work feels more exploratory than methodologically substantial.

**Questions:**

Using code synthesis to model simple, rigid objects is not particularly challenging, as the geometry is relatively easy to capture. What would be more meaningful is demonstrating the ability to model articulated objects that maintain correct articulation and structural relationships. It would also be interesting to examine the capability of VLMs to handle such cases.

---

> ### Author Response · Authors · 2025-11-26
> **Response to Reviewer WYcf**
>
> **Q1: The work feels more exploratory than methodologically substantial.**
> A1: We thank the reviewer for pointing out this concern, but we believe the paper makes concrete methodological contributions beyond being purely exploratory. As shown in Table 1, even when using the strongest model (e.g., Gemini 3.0 Pro), the naive Single-call strategy performs clearly worse than our designed few-shot code reconstruction paradigm.
>
> While planning, RAG, and in-context learning are known ideas, our work integrates those ideas into a unified few-shot scheme tailored for code-based 3D reconstruction. The few-shot exemplars jointly teach (i) how to produce quantitative blueprints and (ii) how to invoke Blender APIs correctly. As a result, this concrete realization allows our code-based pipeline to reach, and sometimes surpass \textit{InstantMesh}, a widely recognized feed-forward mesh based method.
>
> What's more, we introduce and analyze a new paradigm for 3D reconstruction and editing: representing 3D assets as executable Blender code generated by VLMs (3D-CoS). We then systematically quantify the capability frontier of multiple VLMs under this paradigm through extensive experiments on both reconstruction and editing tasks. The pipeline, datasets, and evaluation protocol together form a reusable methodological framework for future work, beyond a one-off exploratory case study.
>
> **Q2: Examine VLMs' ability to model articulated objects.**
> A2: We appreciate the opportunity to demonstrate our method's capability in handling articulated objects. We evaluated the effectiveness of our Blender-based pipeline on samples from three object categories: Cabinet, Monitor, and Toilet. Specifically, we implement the translational motion of cabinet drawers, while applying rotational transformations to achieve the horizontal and vertical swiveling of monitor screens and the axial rotation of toilet lids. All generated articulated objects maintain correct articulation and structural relationships, as shown in Figure 9 in the revised manuscript.

---

### Official Review · Reviewer_J3XK · 2025-11-03

**Soundness:** 2
**Presentation:** 3
**Contribution:** 4
**Rating:** 6
**Confidence:** 3

**Summary:**

This paper proposes a new 3D reconstruction method, by using blender code as unified representation, authors highlight the editability for this representation.

**Strengths:**

1. This paper shows the great potential of using blender code as 3D representation, and directly reconstruct from single image. This is very valuable as it provides special insights into how VLMs understand the spatial informations.

**Weaknesses:**

1. I feel using blender code for 3D reconstruction is somehow a too early exploration, it's still hard for VLMs to have a good spatial understanding, like the scale, angle, curvature. It is not precise enough. In terms of edition, it is somehow not closely connected with reconstruction task, it does show the advantages of using code for a 3D representation (which is not new), but cannot justify why we need to use the code for reconstruction.
2. I feel this would be harder to extend to scene level reconstruction than traditional methods, as current VLMs are still struggling understanding spatial informations.
3. The performance can be largely improved by fine-tuning a model on specific data, as discussed in sec 4.3
4. The code representation is not possible to reconstruct texture

**Questions:**

1. Instead of directly generate blender code from input image, we can also reconstruct using traditional method, then use MeshCoder to convert these representation to blender code. What's the advantage of 3D CoS against this type of method?
2. What is the time and token cost?
3. What is the Failure rate of o3 and gemini?

Comments: In general, I feels it is weird to use blender code for 3D reconstruction, but I really appreciate authors effort in making it work! I feels this is valuable to test the spatial understanding of VLMs, but we will not end up using code for 3D reconstruction. If authors can provide more justification on this besides edition, I would appreciate it!

---

> ### Author Response · Authors · 2025-11-26
> **Response to Reviewer J3XK**
>
> **Q1: Concerns about VLM's spatial understanding and precision (scale, angle, curvature).**
>
> A1: We thank the reviewer for this thoughtful comment. We acknowledge that early VLMs struggled with precise spatial reasoning. However, our extensive experiments reveal a rapid evolution in this capability. As demonstrated in our updated results, the latest model Gemini 3.0 Pro, when prompted with few-shot paradigm, achieve an F-score@5% of 0.8824 and 3D IoU of 0.3083. These metrics outperform the state-of-the-art classical mesh baseline \textit{InstantMesh} (0.8809 / 0.3049).  This suggests that while VLMs might not be pixel-perfect like optimization methods, they are now capable of generating structurally accurate, watertight, and topologically consistent geometry. We believe this is the right time to explore this direction seriously.
>
> **Q2: Why use code for reconstruction?**
>
> A2: We appreciate the opportunity to clarify the unique value of the code paradigm beyond editing. Beyond editability, we argue that code offers fundamental advantages over explicit representations, most notably in structural validity and alignment with foundation models.
>
> Unlike meshes being unstructured collections of triangles, code inherently organizes objects into semantic parts and hierarchies. This structural validity is critical for downstream applications like physics simulation or robotic manipulation where identifying separate moving parts is necessary.
>
> Moreover, as a textual modality, code aligns natively with modern Large Models, allowing us to directly leverage their massive pre-trained reasoning capabilities. This enables us to incorporate extensive 3D knowledge and programming logic into the reconstruction task, without relying on the massive 3D-specific datasets required to train mesh generators.
>
> **Q3: Concerns about extending to scene-level reconstruction.**
>
> A3: While our current work focuses on object-level reconstruction, we respectfully argue that code is actually more suitable for scene-level extension than traditional methods. A scene is essentially a composition of objects. Code naturally supports this via function calls and hierarchical composition (e.g., `place_object(model, pose)`). Former works like \textit{SceneCraft} and \textit{BlenderGym} [1,2] have already demonstrated the efficacy of code for scene layout. Our work provides the fundamental building blocks for such scene-level programs.
>
> **Q4: Why not "Image $\to$ Mesh $\to$ Code (MeshCoder)"?**
>
> A4:  We thank the reviewer to point out this valid alternative. While the "Image $\to$ Mesh $\to$ Code" pipeline is a feasible path, our direct Image $\to$ Code (3D-CoS) approach offers distinct advantages.
>
> Traditional image-to-mesh reconstruction often produces meshes with noise, holes, or non-manifold geometry. Converting such imperfect meshes into code is an ill-posed inverse problem that often leads to error accumulation or failure to capture the underlying logic. Our method bypasses this noisy intermediate stage.
>
> On the other hand, \textit{MeshCoder}[3] relies on training specific "part-to-code" models, which restricts it to predefined categories and primitives. As noted in the \textit{MeshCoder} paper section A.1.2, it requires custom-designed functions for specific shapes like spoons to handle their geometry. In contrast, our method leverages the VLM's semantic understanding to directly generate code for diverse, open-world objects by composing general-purpose Blender APIs, without being limited to a closed set of learned parts.
>
>
> **Q5: Time, token cost, and failure rate.**
>
> A5: We have logged these metrics in our experiments and included detailed tables in Appendix A.2 & A.3 of the revised manuscript.
>
> - Time & Token Cost: As shown in the table below (Table 8 in the revised manuscript), for Gemini 3.0 Pro, the Single-call paradigm is the most economical. RAG increases the cost but delivers higher accuracy. Regarding the Few-shot paradigm, we note that the cost is strictly dependent on the length and complexity of the specific exemplars used, thus a fixed average statistic varies by case and is less meaningful.
>
> | Metric | Single-call | Planning | RAG |
> | --- | --- | --- | --- |
> | Prompt tokens | 1,222 | 9,093 | 11,867 |
> | Completion tokens | 10,885 | 16,989 | 21,752 |
> | Total tokens | 12,107 | 26,082 | 33,619 |
> | Time per call (mins) | 2.27 | 3.17 | 4.00 |
>
> - Failure Rate: We define a "failure" as the failure to generate executable code after 5 auto-correction retries. Under this protocol, closed-source SOTA models exhibit exceptional robustness. o3 and Claude 3.5 Sonnet achieved a 0% failure rate across Single-call and RAG paradigms. Gemini 3.0 Pro similarly maintained a failure rate of ≤ 2%, demonstrating that code generation reliability is no longer a bottleneck for top-tier VLMs.
>
> (Continued on next comment)

---

> > ### Author Response · Authors · 2025-11-26
> > **Response to Reviewer J3XK (cont'd)**
> >
> > **Q6: Fine-tuning improves performance.**
> >
> >
> >
> > A6: We fully agree with the reviewer. Our primary goal in this paper was to delineate the capability frontier of VLMs without specific training. The fact that zero-shot/few-shot VLMs can already rival trained mesh baselines suggests that a fine-tuned "3D-Code-VLM" could indeed revolutionize this field. We have discussed this potential in the Conclusion and Limitations sections.
> >
> >
> >
> > **Q7: Code representation cannot reconstruct texture.**
> >
> >
> >
> > A7: We thank the reviewer for raising this insightful point regarding texture reconstruction. While our current evaluation focuses on geometry to isolate spatial reasoning, we clarify that code representation is capable of handling textures. The Blender Python API supports procedural material generation (via Shader Nodes), which effectively reconstructs natural textures. Furthermore, given that pixel-level image data is unsuitable for direct programmatic representation, texture maps can be generated using existing tools and integrated as assets. With these assets, the VLM can synthesize code that precisely controls UV mapping and material assignment, achieving arbitrary visual effects. We view these as promising directions for future exploration.
> >
> >
> > [1] Xiuyu Yang and Yunze Man and Jun-Kun Chen and Yu-Xiong Wang. SceneCraft: Layout-Guided 3D Scene Generation. NeurIPS2024. arXiv:2410.09049
> >
> > [2] Yunqi Gu and Ian Huang and Jihyeon Je and Guandao Yang and Leonidas Guibas. BlenderGym: Benchmarking Foundational Model Systems for Graphics Editing. CVPR2025. arXiv:2504.01786
> >
> > [3] Bingquan Dai and Li Ray Luo and Qihong Tang and Jie Wang and Xinyu Lian and Hao Xu and Minghan Qin and Xudong Xu and Bo Dai and Haoqian Wang and Zhaoyang Lyu and Jiangmiao Pang. MeshCoder: LLM-Powered Structured Mesh Code Generation from Point Clouds. arXiv:2508.14879

---

### Author Response · Authors · 2025-11-26
**General Response**

We sincerely thank all reviewers for their insightful feedback and constructive suggestions. We are encouraged by the reviewers' appreciation of our proposed 3D-CoS paradigm and the rigorous evaluation protocol.

We have revised our manuscript to address the concerns raised. These revisions are highlighted in blue color. The key revisions and additional experiments are summarized as follows:

- Evaluation of Gemini 3.0 Pro: We have expanded our evaluation to include Gemini 3.0 Pro, the latest closed-source VLM. With this model, our code-based reconstruction achieves a 3D IoU of 0.3083 and an F-score of 0.8824, surpassing the leading supervised mesh baseline, *InstantMesh* (0.3049 / 0.8809). This result demonstrates the promising future of code-based 3D reconstruction and highlights the substantial capability of the latest VLMs for geometric reasoning without specific fine-tuning.

- Introduction of Few-shot Paradigm: In response to suggestions regarding in-context learning, we introduced a Few-shot paradigm, carefully designed for 3D reconstruction, where the VLM is prompted with high-quality exemplar pairs. Experiments show that this strategy further enhances generation quality compared to planning and few-shot strategies.

- Exploration of Visual Refinement: We conducted new experiments on an iterative visual refinement pipeline. We provide a detailed analysis in Appendix A.4, discussing the current limitations of naive refinement compared to planning and few-shot strategies.

- Additional Analyses & Improvements: We have enriched the manuscript by demonstrating articulated object modeling to highlight the code's structural advantages, incorporating a user study to more objectively demonstrate the advantages of edit, and expanding Related Work with foundational citations.

We believe these revisions substantially strengthen the paper and address the reviewers' concerns. We thank the reviewers again for their time and valuable guidance.

---

### Author Response · Authors · 2025-12-03
**Summary of Rebuttal and Revisions for the Area Chair**

We sincerely appreciate the AC's time and effort in handling our submission and thank all reviewers for their time, meticulous review, and constructive comments, which have been instrumental in refining the quality and scope of our paper.

We are especially encouraged by the recognition of our work's value:
- Reviewer **J3XK** acknowledged our paradigm of performing 3D reconstruction using Blender code as the 3D representation.
- Reviewers **WYcf** and **gLSo** appreciated that our proposed technique works effectively and our benchmarking is both valuable and comprehensive.
- Reviewer **n5AR** highlighted the fine-grained control and editing enabled by framing 3D reconstruction as a code generation problem.

We have carefully addressed every identified weakness and question, and have made detailed revisions to the main text and Appendix (all highlighted in blue). The main issues and our responses are summarized below:

**[Additional Experiments] Issue 1: In-context learning and refinement experiments**

Regarding in-context learning, we introduced a few-shot paradigm with carefully designed, high-quality exemplar pairs. Experiments show that this strategy further improves reconstruction quality, demonstrating the effectiveness of in-context guidance for precise 3D code synthesis.

Besides, we designed a refinement paradigm and conducted detailed experiments, which reveal that a simple refinement pipeline does not deliver the expected benefits, indicating that there remains substantial room for future exploration in using VLMs to refine code.

**[Additional Experiments] Issue 2: Maturity of VLM-based reconstruction**

Regarding the maturity of VLM-based reconstruction, we evaluated the performance of Gemini 3.0 Pro on 3D reconstruction tasks. Experiments demonstrate that when integrated with the few-shot strategy, this combination enables off-the-shelf VLMs to achieve geometric reasoning capabilities that rival the mesh-based baseline *InstantMesh*. Based on these results, we believe that a well-designed "3D-Code-VLM" could indeed revolutionize this field, and we view our work as a foundation justifying this direction.

**[Clarification] Issue 3: More of an evaluation paper than a methodological paper**

Our work not only proposes a complete evaluation protocol to thoroughly assess the capabilities of existing VLMs, but also makes concrete methodological contributions.

As shown in Table 1 in the manuscript, even when using the strongest model, the naive Single-call strategy performs clearly worse than our designed few-shot code reconstruction paradigm. While planning, RAG, and in-context learning are known ideas, our work integrates those ideas into a unified few-shot scheme tailored for code-based 3D reconstruction. As a result, this concrete realization makes our code-based pipeline the first to rival the leading supervised mesh-based baseline *InstantMesh*.

**[Clarification] Issue 4: Comparison with *MeshCoder***

We clarified the distinct advantages of our paradigm over *MeshCoder* [1]. We emphasized that *MeshCoder* relies on 3D point cloud inputs, which restricts generalization and introduces error accumulation in image-based tasks with "Image->Mesh->Code" process. In contrast, 3D-CoS operates directly from single RGB images in an open-world setting without 3D-specific training, making it significantly more applicable to real-world scenarios where only 2D observations are available. Besides, *MeshCoder* requires introducing additional structural designs when dealing with objects with special structures (such as scissors), whereas 3D-CoS does not have this limitation.

In addition to the common issues mentioned above, we have provided detailed responses to all specific issues raised by each reviewer, which are not listed here individually. We sincerely appreciate AC's effort in fully considering our detailed rebuttal and clarifications. Thank you again for your dedication.

[1] Bingquan Dai and Li Ray Luo and Qihong Tang and Jie Wang and Xinyu Lian and Hao Xu and Minghan Qin and Xudong Xu and Bo Dai and Haoqian Wang and Zhaoyang Lyu and Jiangmiao Pang. MeshCoder: LLM-Powered Structured Mesh Code Generation from Point Clouds. arXiv:2508.14879

---

### Meta-Review · Area_Chair_xigr · 2026-01-07

**Summary:**

Common concerns from reviewers include:
1. Lack of technical novelty (Reviewer J3XK, WYcf, n5AR).
2. Not obvious to extend to more complex 3D reconstruction setup, such as scene (Revewer J3XK), more complex objects (Reviewer n5AR), or textures (Reviewer J3XK).
3. Need justification of why not exploring other technique such as in-context learning and iterative refinements (e.g. Reviewer n5AR, gLSo).
4. Evaluation could use some improvements. Reviewer n5AR pointed out the user study might be needed; and reviewer gLSo points out a potential baseline: MeshCoder.
There are also concern of clarity of the expositions (e.g. Reviewer gLSo) and reproducibility (Reviewer n5AR)
5. Reviewer J3XK concerns about computational costs.

**Reviewer Concerns:**

The author mainly provide the experiments on in-context learning and iterative refinement. The results show that in-context learning helps and iterative refinement doesn't provide additional benefits. This partially address the reviewers' concerns but also showed that the paper's exploration wasn't comprehensive. The author also provides some profiling of computational costs.

In response to the technical novelty, the authors try to make an argument that the technical contributions include both showing the potential of  the VLLM-based 3D reconstruction pipeline and building a system that leverages different existing techniques to make such pipeline work.

Whether the pipeline can be potentially extended to harder 3D reconstruction is still a hanging question.

**Reviewer Scores:**

Reviewer J3XK might slightly lower the score since the concern about whether it can be extended to scene level might not be addressed well and the computational costs for the provided system is also non-trivial.

Reviewer gLSo's concern about comparing with MeshCoder might still remain. It is not clear to me how few-shot experiments help to address the question gLSo has about the zero-shot set-up. In fact, it could even backfire given that the number shows that in-context learning is performed better. This seemed to suggest a rather major significant deviation from the author's original proposal. With this, I imagine gLSo might remain negative.

Similar rationale might apply to n5AR since the rebuttal suggests that in-context learning, a common techniques, can already lead to better results, which wasn't explored initially and it seems to backfire the suggestions that the original pipeline contains good technical insight.

The main concern of Reviewer WYcf regards the technical contribution. I believe the reviews from gLSo and n5AR share the same concern that isn't addressed by the author's rebuttal.

---

### Decision · Program_Chairs · 2026-01-26

Reject